

# VESPA-22: a ground-based microwave spectrometer for long-term measurements of Polar stratospheric water vapor

Gabriele Mevi[1,2], Giovanni Muscari[1], Pietro Paolo Bertagnolio[1,4], Irene Fiorucci[1], and Giandomenico Pace[3]

[1]Istituto Nazionale di Geofisica e Vulcanologia, Rome, 00143, Italy
[2]Roma Tre University, Mathematics and Physics Department, Rome, 00146, Italy
[3]ENEA, Laboratory for Observations and Analyses of Earth and Climate, Santa Maria di Galeria, 00123, Italy
[4]Now at: AECOM, Croydon, CRO 2AP, United Kingdom

*Correspondence to*: Gabriele Mevi (gabriele.mevi@ingv.it)

**Abstract.** The new ground-based 22 GHz spectrometer, VESPA-22 (water Vapor Emission Spectrometer for Polar Atmosphere at 22 GHz) measures the 22.23 GHz water vapor emission line with a bandwith of 500 MHz and a frequency resolution of 31 kHz. The integration time for a measurement is of the order of hours, depending on season and weather conditions. Water vapor spectra are collected using the beam switching technique. VESPA-22 is designed to operate automatically with minimum need of maintenance; it employs an uncooled front-end characterized by a receiver temperature of about 180 K and its quasi-optical system presents a half power full beam angle of 3.5°. VESPA-22 measures also the sky opacity with a temporal resolution of two measurements an hour using the tipping curve technique. The instrument calibration is performed automatically by a noise diode; the emission temperature of this element is measured two times an hour through the observation of a black body at ambient temperature and of the sky at 60° of elevation. The retrieved profiles obtained inverting a 24-hour integration spectra present a sensitivity higher than 0.8 from about 25 to 72 km of altitude, a vertical resolution from about 12 to 23 km (depending on altitude) and an overall 1σ uncertainty between 5 and 12 %.

In July 2016, VESPA-22 was installed at the THAAO (Thule High Arctic Atmospheric Observatory) located at Thule Air Base (76.5° N, 68.8° W), Greenland, and has been operating almost continuously since then, with very few interruption periods characterized by poor weather. The VESPA-22 water vapor mixing ratio vertical profiles discussed in this work cover the period from July 2016 to May 2017 and are compared with Version 4.2 of concurrent Aura/MLS (Waters et al., 2006) water vapor vertical profiles. In the sensitivity range of VESPA-22 retrievals, the intercomparison between the VESPA-22 dataset and Aura/MLS dataset convolved with VESPA-22 averaging kernels reveals a correlation coefficient of about 0.9 or higher and an average difference reaching its maximum of -6% or -0.2 ppmv at the top of the sensitivity range.



## 1 Introduction

The Polar atmosphere is a very complex system in which water vapor plays an important role. Water vapor has a major impact on radiative balance affecting both infrared radiation by greenhouse effect and visible radiation by clouds coverage. The importance of water vapor in the Arctic region is enhanced by the so called Arctic Amplification effect (Serreze and Francis, 2006), a positive feedback that links the quantity of water vapor in the atmosphere, the presence of clouds, the ice coverage, and the surface temperature. Additionally, about 10% of the surface warming measured during the last two decades can be ascribed to stratospheric water vapor, as shown by Solomon et al. (2010).

The characterization of the water vapor profile, particularly in the mesosphere and stratosphere, is important to understand many chemical processes. Polar water vapor is directly involved in the ozone chemistry as the main source of the OH radical in reactions that cause the ozone destruction (Solomon, 1999). It is also related to the formation of polar stratospheric clouds (PSCs), that grant the catalytic surfaces on which reactions take place.

The main sources of middle atmosphere water vapor are transport through the tropical tropopause and methane oxidation, whereas the main sink is photolysis in the Lyman-band. In the middle atmosphere, the lifetime of this gas varies from several days to weeks. This long lifetime makes its a valuable tracer for the investigation of dynamical processes that characterize the Polar regions, and the Polar vortex in particular.

The processes that lead to long-term variations in stratospheric and mesospheric water vapor are not completely understood. Positive trends were observed during the last two decades (Nedoluha et al., 1999; Rosenlof et al., 2001) and Oltmans et al. (2000) suggested that only one half of these changes are related to anthropogenic activities.

The comprehension of the change in climate that affects the Arctic and the peculiar characteristics of this region's atmosphere calls for long-term measurements. For this task, ground-based microwave remote sensing is a powerful tool to measure water vapor profiles and total amount of precipitable water vapor (PWV).

In this manuscript, VESPA-22 (water Vapor Emission Spectrometer for Polar Atmosphere at 22 GHz), a microwave spectrometer developed at the INGV (Istituto Nazionale di Geofisica e Vulcanologia) is presented. In July 2016, during the Study of the water VApor in the polar AtmosPhere (SVAAP) measurements campaign, a key part of a research effort devoted to the study of the impact of water vapor and clouds on the radiation budget at the ground, VESPA-22 was installed at the Thule High Arctic Atmospheric Observatory (THAAO) located at Thule Air Base (76.5° N, 68.8° W), Greenland. Here the instrument general features, the measurement physics and technique, and a comparison between VESPA-22 and Aura/Microwave Limb Sounder (MLS) (Waters et al., 2006) datasets are presented, with data ranging from July 2016 to May 2017. MLS is an instrument on board of NASA's Aura satellite. Due to its Sun-synchronous polar orbit the satellite overpasses Thule two times a day at fixed local times. For the comparison shown in this manuscript we used version 4.2 (v4.2) MLS data.



## 2 Instrumental setup

VESPA-22 collects the microwave radiation emitted by the water vapor transition at 22.235 GHz with a spectral resolution of 31 kHz and a bandwidth of 500 MHz. From the spectral measurements water vapor profiles can be retrieved with a temporal resolution of 2-4 profiles a day, depending from weather conditions and season. The instrument also measures the

sky opacity with a temporal resolution of few minutes.

In Figure 1 a photo and the layout of the spectrometer are displayed. VESPA-22 collects the sky signal coming from different elevation angles using a parabolic mirror. The signal is reflected by the mirror to a Gaussian choked horn antenna (Teniente et al., 2002). This antenna has a far field directivity of 23.5 dB with a pattern that can be approximated at 99.85% to a Gaussian beam with a beam waist of 22.4 mm (Bertagnolio et al., 2012). The quasi-optical system, antenna and

parabolic mirror, has a full beam at half power $\theta_{3dB} = 3.5°$ (Bertagnolio et al., 2012). This high directivity of the system allows the observation at angles as low as 12° above the horizon, therefore maximizing the number of emitting molecules and the Signal-to-Noise ratio of the measurement. The waveguide used by VESPA-22 polarizes the incoming radiation with a gain difference between the two polarization modes of 45 dB.

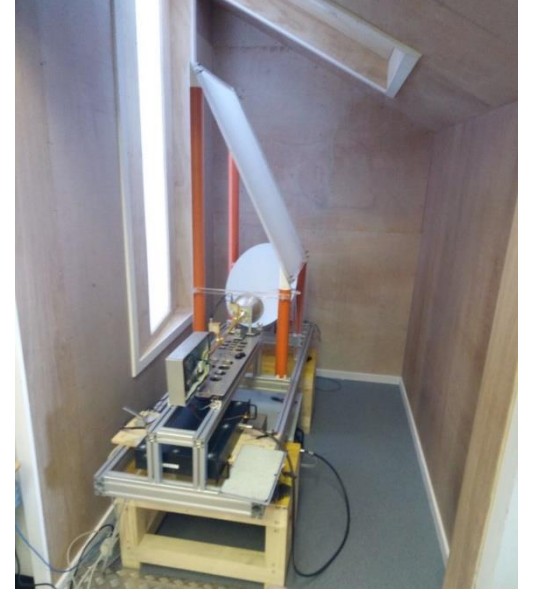

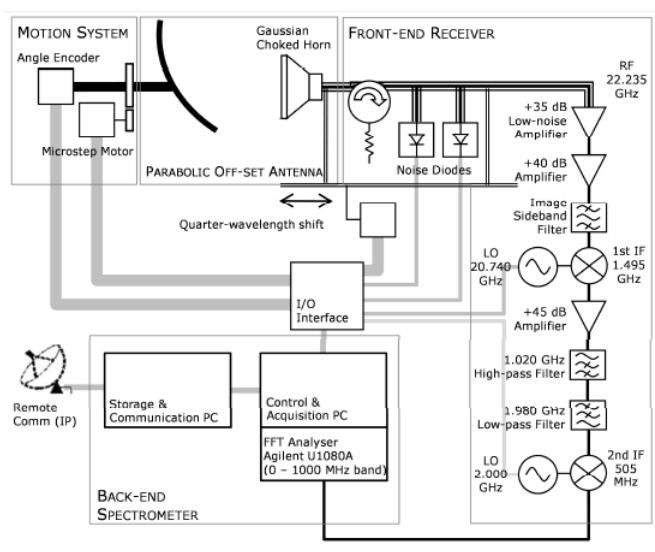

(a)                                              (b)

**Figure 1: (a), a picture of the instrument installed at the THAAO, Thule Air Base, Greenland where the two eccofoam windows can be seen. (b), a scheme of the instrument VESPA-22.**

A first uncooled low noise amplification stage amplifies the incoming signal with a gain of 35 dB. Two noise diodes are inserted in the IF chain after the choke antenna by means of two 20 dB broadwall direction couplers. These two diodes

produce a signal which is measured to be about 119 K and 78 K and are used to calibrate the sky signal observed by VESPA-22, as described in the Sect. 4.2.



After the 20 dB couplers the signal is amplified and down-converted to lower frequencies and eventually sampled at 2Gs/s by a Fast Fourier Transform Spectrometer (FFTS) Acquiris AC240/Agilent U1080A-001 with 16384 channels, resulting in a spectral resolution of 31 kHz for a total bandwidth of 500 MHz.

The receiver temperature $T_{rec}$ of the instrument is measured using a $LN_2$ calibration scheme and has a mean value of 181 K.

During the development of the instrument every effort was made to avoid multiple internal reflections of the signal which would cause the formation of standing waves affecting the observed spectrum. For this reason, during measurements, the antenna is moved back and forth to change its distance from the mirror by $\lambda/4 = 3.34\ mm$. One full spectrum is obtained by averaging together two 6-minute spectra collected with the antenna positioned at these two different distances from the mirror. The destructive interference minimizes the standing waves caused by multiple reflections internal to the system. Two

photodiodes are employed to check and reset daily the correct distance between the antenna and the mirror.

VESPA-22 is installed indoor to preserve it from the strong winds and storms which occur during the Polar winter season. The indoor installation prevents the deposition of snow or dust, as well as the condensation of water droplets, on the quasi-optical system, therefore improving the durability of the equipment. Additionally, the parabolic mirror and its driving motor are not exposed to strong winds, and VESPA-22 is therefore characterized by a pointing offset very stable with time. The

15 spectrometer observes the sky through two 5-cm thick Plastazote LD15 windows, one covering the observation at angles from 10° to 60° above the horizon, and a smaller one covering the zenith direction (showed in Figure 1 and Figure 2). This material was tested in the laboratory and proved to have a negligible absorption in the microwave region of interest. The windows are however not perpendicular with respect to the antenna beam in order to minimize the formation of standing waves. The eccofoam opacity was estimated during VESPA-22 installation and it was measured to be less than 0.0005

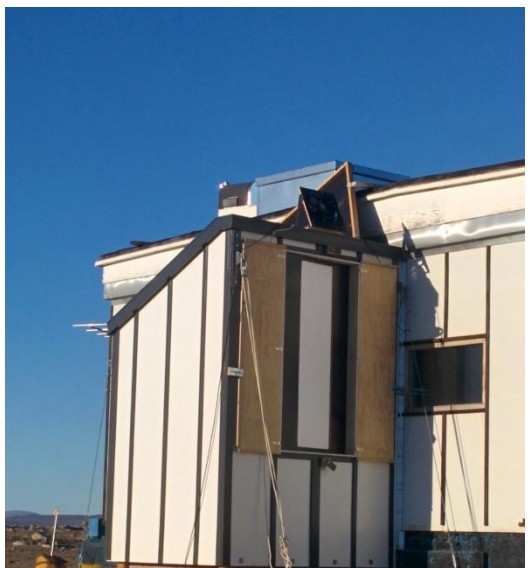

**Figure 2: a photo of the exterior of the wooden annex hosting VESPA-22. The observing window of the signal beam is visible on the side of the annex.**





Nepers. Two powerful fans are utilized to blow off the snow from the observing windows, preventing deposition or ice formation on the windows. The instrument is located in a small wooden annex (Figure 2) to the main observatory in order to minimize the presence of metal surfaces which could produce standing waves.

VESPA-22 compares the sky emission from the zenith direction (also called reference beam) to the sky emission from an angle close to the horizon (also called signal beam) in order to perform a stratospheric measurement, as it will be explained in the Sect. 4.1. The zenith is observed through a Delrin® acetal homopolymer resin sheet, hereafter simply delrin, (see Figure 1 a) that adds a grey body emission to the zenith emission; this element forms an angle with the incident beam equal to the Brewster's angle, in order to minimize the reflection and the formation of standing waves.

During an hour of standard instrument operations, about 3 minutes are dedicated to tipping curves (Sect. 4.3) used to measure tropospheric opacity, about 35 minutes are dedicated to the observation of signal and reference beams, while the rest of the time is dedicated to instrumental operations, such as the rotation of the parabolic mirror or the instrument calibration by means of the noise diodes.

## 3 Measurement physics

VESPA-22 collects the 22.235 GHz radiation emitted by water vapor molecules. At this frequency, the Rayleigh Jeans approximation for the Plank's Law can be used and the radiation intensity can be expressed in terms of brightness Temperature $T_\nu$. This quantity is a function of the frequency ν. The radiation intensity collected at the ground can be described using the radiative transfer equation for the elevation angle θ:

$$T_\nu(z_0) = T_0 e^{-\mu(\theta)\tau_\nu(z_{toa})} + \mu(\theta)\int_{z_0}^{z_{toa}} T(z)\alpha_\nu(z)e^{-\tau_\nu(z)\mu(\theta)}dz \ , \tag{1}$$

where $T_\nu(z_0)$ is the radiation measured by the instrument located at an altitude $z_0$, $T_0$ is the incoming radiation at the top of the atmosphere $z_{toa}$, $\alpha_\nu(z)$ is the absorption coefficient, $T(z)$ is the physical temperature at altitude z, $\tau_\nu(z)$ is the opacity defined as

$$\tau_\nu(z) = \int_{z_0}^{z}\alpha_\nu(s)ds \ , \tag{2}$$

and $\mu(\theta)$ is the air mass factor at the elevation angle θ, computed according to de Zafra (1995).

$$\mu(\theta) = \frac{R + z_{trop}}{\sqrt{\left(R + z_{trop}\right)^2 - \left(\left(R + z_{obs}\right)\cos(\theta)\right)^2}} \ . \tag{3}$$





In the latter expression, R is the Earth's radius, $z_{trop} = 3\ km$ is the width of the tropospheric layer in which most of the atmospheric water vapor is present, and $z_{obs}$ is the altitude of the instrument (0.2 km).

In the stratosphere, the line shape of the emission from an altitude z is mainly a function of atmospheric pressure

$$\Delta \nu \propto P(z)\left(\frac{T_0}{T(z)}\right)^x ,\qquad (4)$$

where $\Delta \nu$ is the line width, $x < 1$ is a constant coefficient and $T_0 = 300\ K$. Using this dependence and knowing the pressure and temperature atmospheric profiles, the measured spectrum can be inverted using the reverse problem theory to obtain the vertical water vapor mixing ratio profile.

In the mesosphere, the Doppler broadening overcomes the pressure broadening determining an upper limit to the altitude range in which the water vapor profile can be retrieved. In the lower stratosphere, the broadening makes the line width comparable to VEPSA-22 spectral bandwidth setting a lower altitude limit for the deconvolved profile.

## 4 Measurement technique

The major part of the incoming radiation measured by VESPA-22 comes from the troposphere. The tropospheric emission needs to be filtered in order to evidence the stratospheric signal.

In order to simplify Eq. (1), the following approximations can be adopted (e.g., Nedoluha et al., 1995).

- The troposphere is represented as an isothermal layer absorbing the signal to be measured. The first kilometers of the atmosphere produce the greatest part of this layer emission which has a line width larger than VESPA-22 bandwidth. This contribution is treated as an emission constant in frequency.
- The contribution of the stratospheric water vapor absorption to the opacity $\tau$ is negligible. This approximation was tested by calculating the atmospheric opacity by means of the radiative transfer simulation software ARTS (Eriksson et al., 2011) and using water vapor vertical profiles with and without their stratospheric component. At the frequency of maximum absorption, the contribution of the stratospheric profile to the overall atmospheric opacity in the 22 GHz region is between 2% and 5% depending from the season. When averaged over the VESPA-22 frequency range, the difference between the opacity calculated using a normal vertical water vapor profile and a profile with no water vapor above the tropopause is between 0.4% and 1.5% depending from the season. In Figure 3, the opacity calculated by ARTS using complete water vapor profiles from the ground to 110 km altitude and the opacity calculated with the same profiles but no water vapor above the tropopause are compared. The profile used were measured by MLS/Aura above Thule on different seasons.
- The opacity $\tau_\nu$ can be substituted by its mean value $\tau$. The maximum difference between $\tau_\nu$ and its mean value $\tau$ is between 1.6 and 3.6 % depending from the season.
- The only signal coming from outside the atmosphere, $T_0$, is the cosmic background radiation with a constant brightness temperature of 2.73 K.





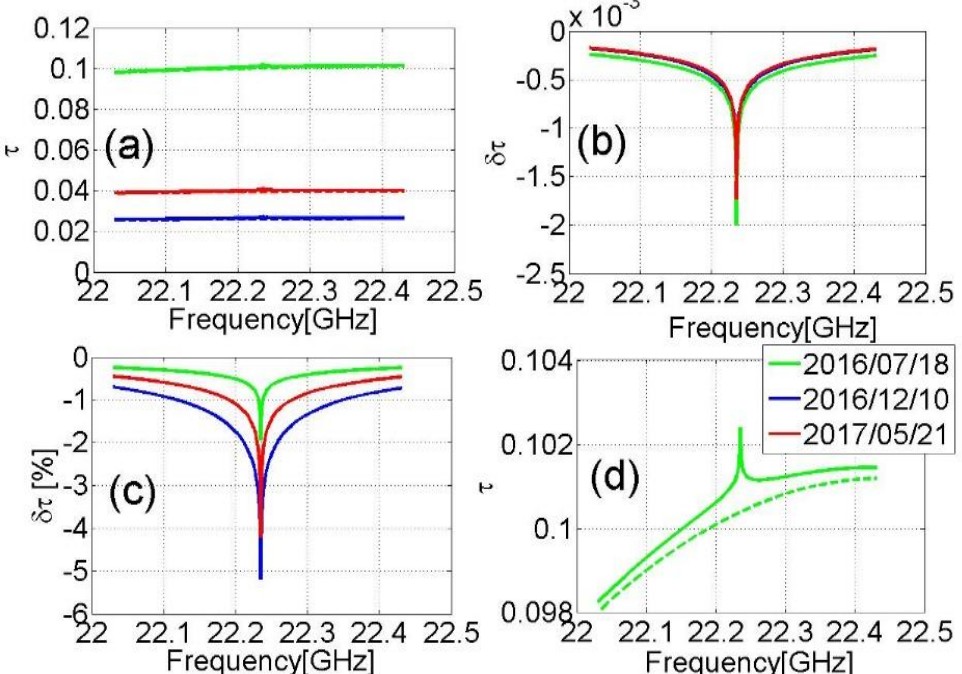

**Figure 3: (a) atmospheric opacity calculated by ARTS for complete water vapor profiles measured by AURA/MLS above Thule region on different seasons (solid lines) and for the same profiles with no water vapor above 10 km (dashed lines); (b) and (c) absolute and relative difference between the tropospheric and total opacity; (d) particular of the opacity of the 2016/07/18 profile.**

With these approximations Eq. (1) can be written as

$$T_S = T_0 e^{-\mu\tau} + T_{atm}\left(1-e^{-\mu\tau}\right) + \mu T(\nu) e^{-\mu\tau} \ . \tag{5}$$

The left term is the radiation received by the spectrometer from the elevation angle θ, the first term in the right side is the extra-atmospheric emission. The second term in right side is the solution of the radiative transfer equation for an isothermal domain and it represents the tropospheric emission with $T_{atm}$ as the mean troposphere temperature weighted with the water

vapor concentration. The third term is the emission coming from the stratosphere and mesosphere, $T(\nu)$, proportional to the air mass factor $\mu$ and attenuated by the troposphere. In all the previous terms $e^{-\mu\tau}$ is the tropospheric signal absorption. The third term on the right side approximates the integral term of Eq. (1), according to Eq. (6) and (7)

$$\mu \int_{z_0}^{z_{toa}} T_\nu(z) e^{-\tau_\nu(z)\mu(\theta)} dz \sim \mu e^{-\tau\mu} \int T_\nu(z) dz \ \text{and} \tag{6}$$

$$T(\nu) = \int T_\nu(z) dz \ , \tag{7}$$

where it was used the relation between physical and brightness temperature in the Rayleigh Jeans approximation $T_\nu = \alpha_\nu T$ .

$T(\nu)$ is the stratospheric signal that has to be inverted in order to retrieve the water vapor stratospheric profile.



## 4.1 The balancing beams technique

The technique that VESPA-22 employs to measure the stratospheric signal is called balancing-beam technique or Dicke switching technique (e.g., Parrish et al., 1988). The instrument compares the emission coming from an observation angle θ

5 (signal beam) with a reference signal with the same mean power over the passband. The observation angle depends from atmospheric opacity and for VESPA-22; it varies from 12° to 25° above the horizon. The reference signal used by VESPA-22  is the sky emission at the zenith (reference beam). In clear sky conditions the emission at the zenith is smaller than the emission at a much larger zenith angle. Therefore, in order to ensure that the reference beam has the same mean power of the signal beam, a thin sheet of delrin is inserted in the reference beam. The delrin sheet acts as a grey body so that:

$$T_R = T_0 e^{-\tau-\tau_d} + T_{atm}\left(1-e^{-\tau}\right)e^{-\tau_d} + T(\nu)e^{-\tau-\tau_d} + T_d\left(1-e^{-\tau_d}\right), \tag{8}$$

where $T_R$ is the radiation observed by VESPA-22 coming from the zenith, partially absorbed and reemitted by the delrin sheet, $T_d$ is the physical temperature of the sheet and $\tau_d$ its opacity. In this equation, the air mass factor $\mu$ is equal to 1. During data taking operations, VESPA-22 alternates zenith and signal angle observations. The instrument constantly checks if the two beams have the same mean power and changes the signal angle to minimize the difference between them. When

15 the two beams have the same intensity, the frequency independent terms of Eq. (8) and Eq. (5) can be equated obtaining:

$$T_0 e^{-\tau-\tau_d} + T_{atm}\left(1-e^{-\tau}\right)e^{-\tau_d} + T_d\left(1-e^{-\tau_d}\right) \approx T_0 e^{-\mu\tau} + T_{atm}\left(1-e^{-\mu\tau}\right), \tag{9}$$

where the stratospheric contribution to the mean beam intensity (about 1%) is neglected (de Zafra, 1995).

The stratospheric signal $T(\nu)$ is obtained by subtracting the signal and the reference beams. Using Eq. (5), (8) and (9) it is possible to write:

$$T_S - T_R \approx T(\nu)\left(\mu e^{-\mu\tau} - e^{-\tau-\tau_d}\right) \text{ and} \tag{10}$$

$$T(\nu) = \frac{T_S - T_R}{\mu e^{-\mu\tau} - e^{-\tau-\tau_d}} . \tag{11}$$

Three delrin sheets with different thickness (3, 5 and 9 mm) and opacity can be employed, depending from season, in order to maintain the observation angle between 12° and 25° above the horizon. Spectra collected are smoothed using a 50-channel moving average. This smoothing process is not performed in a 6 MHz interval centered around the emission line to maintain

25 the maximum frequency resolution near the peak.

## 4.2 Calibration method

The broad-band response of VESPA-22 to the signal coming from the sky can be written as:

$$V = \alpha\left(T + T_{rec}\right) + V_0 . \tag{12}$$





In this equation, V is the incoming signal in counts number of the FFT back-end spectrometer, T is the signal brightness temperature, $\alpha$ the gain of the instrument, $T_{rec}$ the receiver noise temperature and $V_0$ the "zero" signal of the FFTS. All the quantities represented in Eq. (12) are frequency dependent. The "zero" signal is measured and subtracted to every acquired spectrum so it is not explicitly written in what follows in order to simplify the notation.

VESPA-22 collects the sky radiation from signal and reference beams and subtracts the counts number from these two sources. Using Eq. (12), Eq. (11) can be written as:

$$T(v) = \frac{1}{\alpha} \frac{V_S - V_R}{\mu e^{-\mu\tau} - e^{-\tau - \tau_d}} \ .$$ (13)

VESPA-22 measures the gain parameter $\alpha$ using a noise diode manufactured by Noisecom, an element producing a stable signal and inserted between the choked horn antenna and the first stage low noise amplifier by means of a 20 dB broadwall
coupler (see Figure 1 b). As mentioned before, VESPA-22 has two different noise diodes, one used to perform regular calibrations and the second to ensure the stability over time of the first one, as described by Gomez et al. (2012).

During measurements, the calibration noise diode is switched on and its emission is added to the reference signal. The noise diode is then switched off and VESPA-22 measures only the radiation coming from the zenith. Therefore, the received radiation is:

$$V_{R+ND} = \alpha\left(T_R + T_{ND} + T_{rec}\right) \text{ and }$$ (14)

$$V_R = \alpha\left(T_R + T_{rec}\right) \ .$$ (15)

$V_{R+ND}$ and $V_R$ are the signals expressed in counts number measured with the noise diode turned on and off, respectively, and $T_{ND}$ the noise diode emission temperature. The gain parameter can be obtained by subtracting these two measurements

$$\alpha = \frac{V_{R+ND} - V_R}{T_{ND}} \ ,$$ (16)

where $T_{ND}$ can be estimated by performing a calibration procedure. The calibration consists in measuring the emissions from two sources at two different known emission temperatures. The first source is a black body at ambient temperature made with an eccosorb CV-3 panel by Emerson and Cuming; the second source is the sky at an angle of 60° above the horizon. In the calibration procedure, the sky can be replaced by a second eccosorb CV-3 panel immersed in liquid nitrogen (LN$_2$). The use of liquid nitrogen likely grants more accurate results, as the physical temperature of the emitting body has a smaller
uncertainty with respect to the estimated sky temperature at 60°, but the LN$_2$ calibration cannot be performed automatically by the instrument and so far has been performed in July and November 2016, and February 2017.

The general calibration equations are described in what follows, whereas the tipping curve procedure that allows the use of the sky as calibration source is explained in Sect. 4.3.

Knowing the emission temperature of two sources $\alpha$, $T_{rec}$ and $T_{ND}$ can be obtained with:

$$\alpha = \frac{V_{hot} - V_{cold}}{T_{hot} - T_{cold}} \ , \tag{17}$$

$$T_{rec} = \frac{T_{hot} V_{cold} - T_{cold} V_{hot}}{V_{hot} - V_{cold}} \ , \text{ and} \tag{18}$$

$$T_{ND} = \frac{V_{cold+ND} - V_{cold}}{\alpha} \ . \tag{19}$$

$T_{hot}$ and $T_{cold}$, and $V_{hot}$ and $V_{cold}$ are the emission temperatures and the recorded counts number of the two sources,
respectively. For the black body the emission temperature is assumed to be equal to its physical temperature.

$T_{ND}$ and all the quantities used in Eq. (17), (18), and (19) are spectral quantities. The noise diode produces a signal that can
be considered constant in frequency within 1.5%. The spectra originated from black body measurements (especially those
from the CV-3 immersed in LN$_2$) can be affected by standing waves and in order to avoid to transfer them in the calibrated
sky spectral measurements, $T_{nd}$ is averaged over the central 11000 channels of the FFT spectrometer, as suggested by
Gomez et al. (2012). Therefore:

$$T_{ND} = \left( T_{hot} - T_{cold} \right) \sum_{i=3000}^{14000} \frac{V_{cold+ND}(i) - V_{cold}(i)}{V_{hot}(i) - V_{cold}(i)} \ , \text{ where} \tag{20}$$

$V_{cold+ND}(i)$ is the value of channel $i$ with noise diode turned on while $V_{cold}(i)$ is the value of the same channel with the
noise diode turned off.

### 4.3 Tipping curve calibration

In this section, the tipping curve calibration technique employed to calibrate the noise diodes using the sky signal is
described. The tipping curve procedure is performed twice every hour as it is also used to measure the atmospheric opacity
of Eq. (13). During a tipping curve VESPA-22 collects the radiation coming from different elevation angles, approximately
every 5° from 35° to 60° above the horizon. The measured spectra are averaged using the 11000 central channels of the
spectrometer. Radiation from the stratosphere contributes less than 1% and can be neglected, so the signal intensity can
describe by means of the following:

$$\overline{T}(\theta_i) \cong T_0 e^{-\mu(\theta_i)\tau} + T_{atm}\left(1 - e^{-\mu(\theta_i)\tau}\right) \ . \tag{21}$$

The atmospheric temperature $T_{atm}$ can be estimated from the surface temperature $T_{sup}$ :

$$T_{atm} = T_{sup} - d \tag{22}$$

where the value of $d$ can be affected by seasonal variations. In order to characterize this parameter several radiosoundings
were launched during July, November and December 2016, and February 2017. The value of the parameter $d$ as a function





of time is obtained from a linear interpolation between the mean values of $T_{sup} - T_{atm}$ measured during these four periods, as described in Table 1. $T_{atm}$ is the average temperature obtained from radiosonde data by weighting the tropospheric temperature vertical profile with the water vapor concentration profile.

**Table 1: mean values and standard deviation of $T_{sup} - T_{atm}$ obtained from the radiosoundings**

| Month | Mean $(T_{sup} - T_{atm})$ |
|---|---|
| July | 14.4±2.8 K |
| November | 8.3±3.6 K |
| December | 9.4 ±3.8 K |
| February | 9.4±2.1 K |

Using Eq. (21), it is possible to explicit the relation between the opacity and the mean brightness temperature of the received signal:

$$\mu(\theta_i)\tau = \ln\left(\frac{T_0 - T_{atm}}{\overline{T}(\theta_i) - T_{atm}}\right). \tag{23}$$

A linear regression between the opacities at $\theta_i$, $ln\left(\frac{T_0 - T_{atm}}{\overline{T}(\theta_i) - T_{atm}}\right)$, and the air mass factors $\mu(\theta_i)$ allows us to retrieve the opacity at the zenith, $\tau$ (de Zafra, 1995). Substituting for $\overline{T}(\theta_i)$ using Eq. (12), the Eq. (23) can be written as:

$$\mu(\theta_i)\tau = \ln\left(\frac{T_0 - T_{atm}}{\frac{\overline{V}(\theta_i)}{\alpha} - T_{rec} - T_{atm}}\right) \tag{24}$$

where it appears that $T_{rec}$ and $\alpha$ are needed to perform the calculation. In order to obtain an estimate of these two parameters, during the tipping curve procedure VESPA-22 measures also the emission from a CV-3 eccofoam sheet,
considered as a black body at ambient temperature (hot load).

The sky signal at an elevation angle of 60° ($T_{cold}^{sky}$) acts as second calibration source (cold load). The emission from these two sources is used to calculate $\alpha$ and $T_{rec}$ according to Eq. (17) and Eq. (18). However, since the sky emission temperature $T_{cold}^{sky}$ is not known, VESPA-22 uses an iterative procedure to obtain both $\tau$ and $T_{cold}^{sky}$. An initial opacity value, $\tau_0$, is used as first guess to obtain $T_{cold,0}^{sky}$ using Eq. (21) with $\theta = 60°$. $T_{cold,0}^{sky}$ is then used to obtain $\alpha_0$ and $T_{rec_0}$ from Eq. (17) and Eq. (18);
$\mu(\theta_i)\tau$ is calculated for different elevation angles using Eq. (24), and ultimately a linear fit allow us to calculate a new estimate for $\tau$, $\tau_1$. The iterative procedure goes on until the intercept value is minimized.



The value of $T_{cold}^{sky}$ measured with this procedure is used in Eq. (17) and Eq. (19) to estimate $T_{nd}$. In order to avoid the use of data measured during inhomogeneous-sky condition all measurements producing fits with a root mean square higher than 0.8 are discarded. Figure 4 shows the time series of both noise diodes emission temperatures (blue and cyan) from July 2016 to May 2017. The noise diode in blue is the one used as calibration diode. In the same plot, $T_{nd}$ values obtained using a LN$_2$

cooled eccofoam CV-3 as the cold source are also depicted (orange and red stars). The mean relative difference between $T_{nd}$ values calculated with the two calibration schemes (tipping curve and LN$_2$) is (0.4±0.4)% and (0.2±0.3)% for the calibration and the backup diodes, respectively.

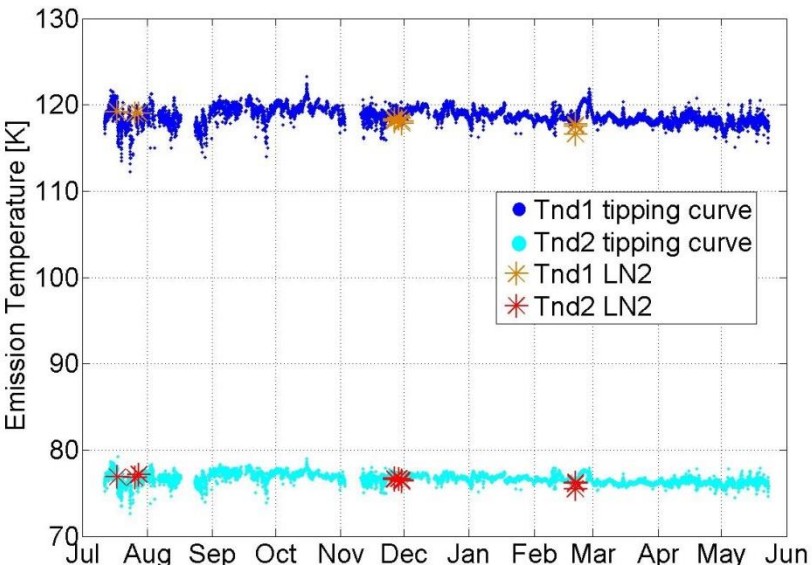

**Figure 4: Time series of the noise diodes emission temperature calculated by means of the tipping curve procedure (blue and cyan**
**full circles) compared with values obtained using a LN2 calibrations (red and orange stars).**

## 5 Retrieval Description

VESPA-22 water vapor vertical profiles are obtained using the optimal estimation theory (Rodger, 2000). The generic relation between the measured spectrum $y$ and the real profile $\tilde{x}$ can be modeled by:

$$y = f\left(\tilde{x}\right) .$$  (25)

Eq. (25) can be expanded to the first order around an apriori profile obtaining the forward equation:

$$y = y_a + K\left(\tilde{x} - x_a\right) ,$$  (26)



where $\tilde{x}$ is the true water vapor atmospheric profile, $x_a$ is the water vapor apriori profile, $y_a$ is the spectrum associated to the apriori vertical profile, and $K$ is the weighting function matrix defined as

$$K = \frac{\partial y}{\partial x}\bigg|_{x_a} . \tag{27}$$

According to Rodgers (2000), the profile $x$ which best represents the atmospheric state $\tilde{x}$ given the measurements $y$ and local climatology $x_a$ and $y_a$ can be retrieved by using

$$x = x_a + G\left(y - y_a\right) , \tag{28}$$

where $G$ is the gain matrix and is defined by

$$G = \left(K^T S_e^{-1} K + S_a^{-1}\right)^{-1} K^{-1} S_e^{-1} . \tag{29}$$

$S_e$ and $S_a$ are the covariance matrices associated to the measurement and the apriori, respectively. $S_a$ contains information on the uncertainties of the apriori, whereas $S_e$ is related to the measurement spectral noise.

The profile $x$ can be used to calculate the synthetic spectrum $y_{fit}$:

$$y_{fit} = y_a + K\left(x - x_a\right) . \tag{30}$$

The altitude grid used for VESPA-22 retrievals starts from 10 km and goes up to 110 km altitude, at steps of 1 km. This range is much larger than the sensitivity interval of the instrument which is in fact limited by the Doppler broadening at high altitudes and by the FFTS bandwidth and the tropospheric influence on the lower stratosphere at low altitudes.

Just the central 400 MHz of the measured spectrum are used in the retrieval.

The $S_a$ matrix is computed according to:

$$S_{a,ij} = \sigma_i \sigma_j e^{-\frac{|z_i - z_j|}{h}} , \tag{31}$$

where $\sigma_i$ and $z_i$ are respectively the root square of the variance (expressed in volume mixing ratio, or vmr) and altitude of the profile showed in Figure 5, while $h$ is a correlation altitude set to be 5 km.



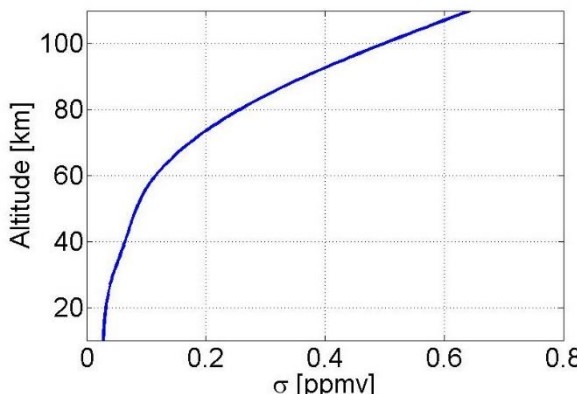

**Figure 5: the value of $\sigma_i$ used to compute the apriori covariance matrix in Eq. (31) as function of the altitude.**

$S_e$ is a diagonal matrix with its diagonal elements constant and calculated using a two-step process. A first retrieval is performed using a fixed value (1x10$^{-5}$) for the $S_e$ diagonal elements and the obtained profile, $x_0$, is used to calculate a

synthetic spectrum $y_{0,fit}$ by means of Eq. (30). In order to consider the spectral measurement noise in the retrieval process, a

second and final inversion is then performed, this time with the $S_e$ diagonal elements set to the $\left( y - y_{0,fit} \right)^2$ mean value.

A second-order polynomial is also added to the retrieval in order to take into account the spectral emission from the upper troposphere and lower stratosphere that would not otherwise be properly resolved by the retrieval algorithm, leaving a spectral baseline unaccounted for (see light blue curve in Figure 6 a).

An important quantity used to characterize the retrieval quality is the averaging kernels matrix $A$ defined as:

$$A = G K \ . \tag{32}$$

The rows of $A$ are called averaging kernels (AKs) and represent the sensitivity of the water vapor retrieval at a given altitude to variations in the water vapor concentration profile at all altitudes (Rodger, 2000). If the AKs are well-peaked functions, centered at their nominal altitude, a perturbation in the atmospheric water vapor concentration at a specific altitude is

transferred by the algorithm to the correct altitude layer of the retrieved profile. Furthermore, the area enclosed under each AK is an indication of the total sensitivity of the retrieved profile at that altitude to atmospheric variations in water vapor concentration. Sensitivity values close to 1 at certain altitudes indicate that the major contribution to the retrieval values at those altitudes comes from spectral measurements rather than from the apriori water vapor profile. Following the suggestion of Tschanz et al. (2013) the retrieval sensitivity range adopted here is the altitude range in which the sensitivity is above 0.8.

The AKs full width at half maximum (FWHM) can instead be used as a rough estimate of the local vertical resolution of the obtained water vapor mixing ratio vertical profile.



## 5.1 Forward model and apriori profile

In order to account for the variations of the mean atmospheric state during the eleven months of data presented in this study, a different water vapor apriori profile is used for each month of measurements. These profiles are obtained from the monthly averages of AURA/MLS water vapor vertical profiles (version 4.2) from the years 2014, 2015, and 2016. These monthly averages are "assigned" to the 15th day of each month and then, at each altitude, are linearly interpolated to build daily apriori profiles that vary gradually, day by day, from the 15th of one month to the next (hereafter this averaging and interpolating process is indicated as "daily smoothing monthly averages"). In the altitude region where MLS data have no scientific relevance, from 85 to 110 km altitude, the apriori profile slowly decreases with altitude as the tail of a gaussian distribution.

The matrix $K$ and the $y_a$ spectrum of Eq. (26) are calculated using the radiative transfer simulation software ARTS (Eriksson et al., 2011), adopting a Voigt-Kuntz lineshape and the line intensity provided by the JPL 2012 catalogue (Pickett et al., 1998, reference site https://spec.jpl.nasa.gov/). Following the work of Selee (1999) and Tschanz et al. (2013), the line described by the JPL 2012 catalogue is divided into three emission lines indicating the hyperfine splitting of the 22.235 GHz water vapor line. The employed pressure broadening and self-broadening parameters are those reported by Liebe (1989). Table 2 summarizes the spectroscopic parameters used for the analysis of VESPA-22 spectral measurements.

**Table 2: Spectroscopic parameters used in VESPA-22 retrieval. Indicated parameters, from left to right, are: emission line frequency, intensity, lower state energy, pressure broadening parameter, pressure broadening temperature dependence, self-broadening, and self-broadening temperature dependence. The line intensity is given for a reference temperature of 296 K.**

| $\nu_0$ [GHz] | S [m² Hz] | E [J] | $\gamma_{air}$ [Hz Pa⁻¹] | $n_{air}$ | $\gamma_{self}$ [Hz Pa⁻¹] | $n_{self}$ |
|---|---|---|---|---|---|---|
| 22.235043990 | 5.3648 10⁻¹⁹ | 8.869693 10⁻²¹ | 28110 | 0.69 | 134928 | 1 |
| 22.235077056 | 4.5703 10⁻¹⁹ | 8.869693 10⁻²¹ | 28110 | 0.69 | 134928 | 1 |
| 22.235120358 | 3.9740 10⁻¹⁹ | 8.869693 10⁻²¹ | 28110 | 0.69 | 134928 | 1 |

The profile $x_{ARTS}$ is the profile used in the forward model calculations to compute the apriori spectrum and the weighting functions. $x_{ARTS}$ matches the apriori profile from 12 km of altitude upward and, below 9 km, it is consistent with the measurements of precipitable water vapor (PWV) collected by the HATPRO radiometer (Rose and Czekala, 2009; Pace et al., 2015) installed at the THAAO. This lower part of $x_{ARTS}$ is calculated according to:

$$x_{ARTS}\left(from\,the\,ground\,to\,9km\right) = \frac{PWV_{Hatpro}}{PWV_{Eu}} x_{Eu} \tag{33}$$

where $x_{Eu}$ is a water vapor mixing ratio profile obtained by daily smoothing monthly averages calculated from radiosoundings launched at the Eureka station (80.0°N -85.9°W)), Canada, $PWV_{Eu}$ is the associated water vapor column




content, and $PWV_{Hatpro}$ is the column content measured by the HATPRO. $x_{ARTS}$ therefore represents the monthly average $x_{Eu}$ profile simply rescaled to be consistent with the column content measurements of the HATPRO at Thule. Data from Eureka were chosen (instead of those from Alert, Canada, for example) because they show the closest resemblance to the tropospheric profiles measured at Thule by local radiosoundings, when the latter are available. In order to avoid

discontinuities in $x_{ARTS}$, values at altitudes between 9 and 12 km are obtained with a linear interpolation between $x_{ARTS}(9\ km)$ and $x_{ARTS}(12\ km)$.

The pressure and temperature profiles needed to run the forward calculation are built merging NASA Goddard Space Flight Center (GSFC), AURA/MLS and climatological temperature and pressure profiles. The NASA GSFC profiles by Goddard Automailer Service (Lait et al., 2005) are used to build the tropospheric meteorological state, from the ground up to 9 km of

altitude. For the altitude range between 10 and 87 km of altitude, the MLS temperature and pressure profiles collected during VESPA-22 observations, in a radius of 300 km from the observation point of VESPA-22, are averaged together to produce a single set of daily meteorological vertical profiles. The VESPA-22 observation point coordinates are chosen to be 74.8° N and 73.5° W, and represent an estimate of the geographical coordinates of the air mass that is observed by VESPA-22 (which points South-West, at about 220°) at 60 km altitude when the instrument aims at an elevation of 15° above the

horizon. Daily temperature and pressure profiles from 86 to 110 km of altitude are obtained by daily smoothing zonal monthly averages from the COSPAR International Reference Atmosphere (Rees et al., 1990).

The absence of vertical discontinuities in the temperature and pressure daily profiles is assured by a smoothing process performed at the altitudes where the three different datasets (GSFC, MLS, and climatological profiles) are stitched together, between 9 and 12 km and between 87 and 97 km.

The software ARTS is employed to simulate the emission from the zenith, $\tilde{y}_r$, and from an angle close to the horizon, $y_s$. The emission $\tilde{y}_r$ is then rescaled using Eq. (34), therefore simulating the effect of the compensating sheet.

$$y_r = \tilde{y}_r e^{-\tau_d} + T_d \left(1 - e^{-\tau_d}\right) \tag{34}$$

The opacity $\tau_d$ used in Eq. (34) is the opacity of the compensating sheet, whereas the temperature $T_d$ is the temperature of the sheet, measured by a sensor installed next to it. The value of the signal beam observing angle imposed in the simulation,

$\tilde{\theta}$, is chosen in order to minimize the mean difference $y_s - y_r$, as it is in fact attained by VESPA-22 in its data taking process. The apriori spectrum is calculated according to the same equation used for the measured signal (Eq. (10) and (11)).

$$y_s - y_r = y_a \left( \tilde{\mu} e^{-\mu \tilde{\tau}} - e^{-\tilde{\tau} - \tau_d} \right) \tag{35}$$

$$y_a = \frac{y_s - y_r}{\tilde{\mu} e^{-\mu \tilde{\tau}} - e^{-\tilde{\tau} - \tau_d}} \tag{36}$$

$\tilde{\mu}$ and $\tilde{\tau}$ are the air mass factor associated to the simulated signal beam and the opacity calculated from the $x_{ARTS}$ profile.

Deriving Eq. (36) and using Eq. (27) and Eq. (34), the retrieval weighting function matrix can be obtained as:





$$K = \frac{\partial}{\partial x}\left(\frac{y_s - y_r}{\tilde{\mu}e^{-\mu\tilde{\tau}} - e^{-\tilde{\tau}-\tau_d}}\right)_{x_a} \cong \frac{\left.\frac{\partial y_s}{\partial x}\right|_{x_a} - \left.\frac{\partial y_r}{\partial x}\right|_{x_a}}{\tilde{\mu}e^{-\mu\tilde{\tau}} - e^{-\tilde{\tau}-\tau_d}} = \frac{K_s - K_r e^{-\tau_d}}{\tilde{\mu}e^{-\mu\tilde{\tau}} - e^{-\tilde{\tau}-\tau_d}} \qquad (37)$$

where $K_s$ and $K_r$ are the weighting function matrices that ARTS calculates for the simulated signal and reference beams. As a first approximation, the dependence of $\tilde{\tau}$ on the stratospheric water vapor profile in Eq. (37) is neglected.

**5.2 Retrieval example**

Figure 6 a shows a VESPA-22 spectrum integrated for second-degree polynomial 24 hours (blue) on 23 December 2016, its corresponding synthetic spectrum $y_{fit}$ (red) and the apriori spectrum (green), while the residual (defined as the difference between fit and measured spectrum, $y_{fit} - y$) is plotted in Figure 6 b. The cyan line is the retrieved by the inversion algorithm. Figure 7 shows the result of the inversion of the measured spectrum depicted in Figure 6 with the apriori profile

10  and uncertainty. The details about uncertainty calculation are discussed in the Sect. 6.

Figure 8 shows the previous retrieval averaging kernels (black and colored solid lines), multiplied by a factor of 10, and the sensitivity (red). In a typical VESPA-22 retrieval the sensitivity is above 0.8 in a range from 26 to 72 km.

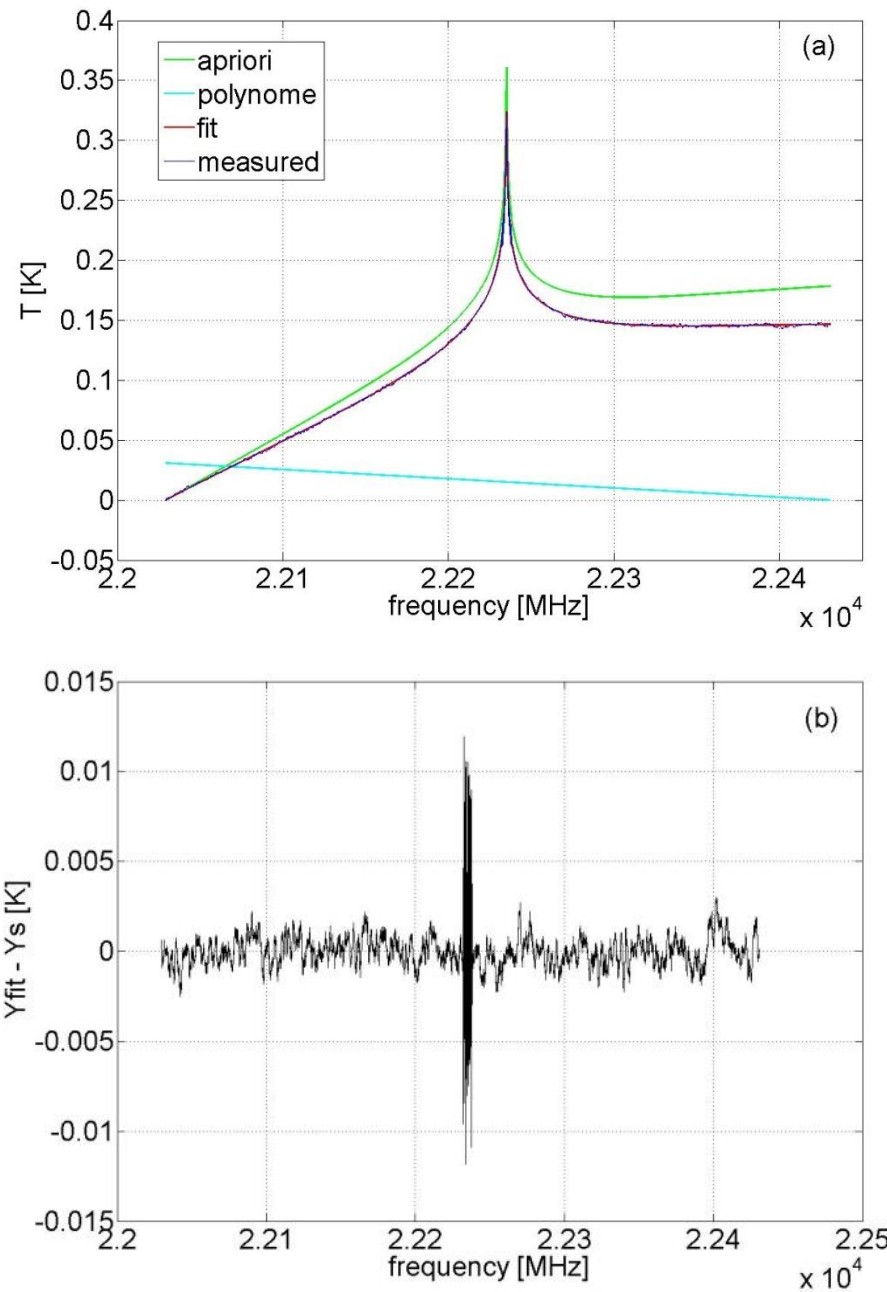

**Figure 6: (a): an example of VESPA-22 measured spectrum (blue) collected on 23 December 2016, the apriori spectrum (green) and the fit spectrum (red); (b): the residual $y_{fit} - y$. The central part of the spectrum is unsmoothed in order to maintain the maximum spectral resolution near the peak and its residual is higher.**



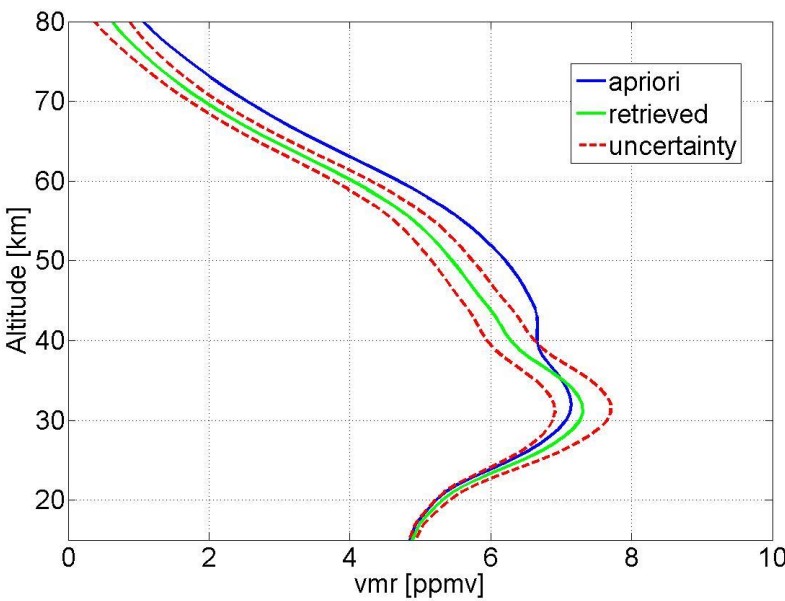

**Figure 7: the retrieved VESPA-22 profile (green solid line) correspondent to the spectrum showed in Figure 6 and the apriori profile (blue solid line). The two red dashed lines describe the uncertainty of this VESPA-22 retrieval (for details on the estimated uncertainty on VESPA-22 mixing ratio vertical profiles see the Sect. 6).**

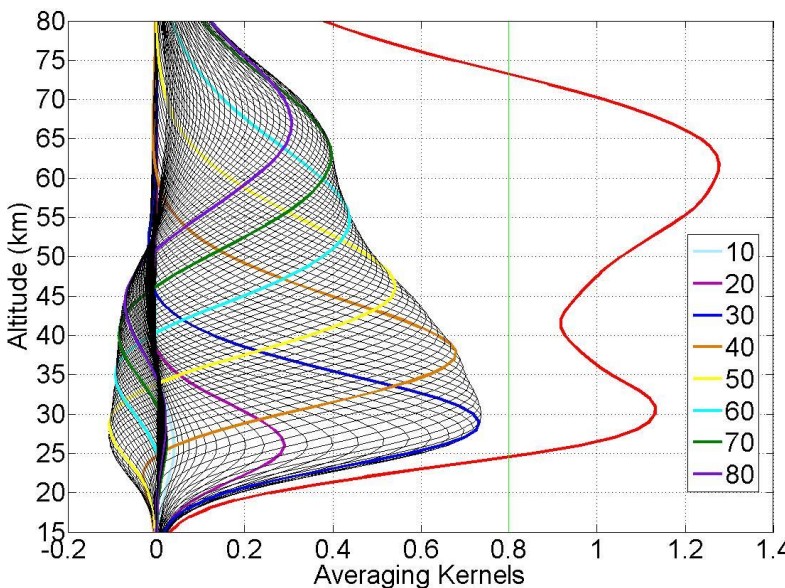

**Figure 8: Rows of the A matrix multiplied by a factor of 10 as a function of altitude (some A functions are highlighted in color). The vertical profile of the sensitivity is shown in red.**



## 6 Retrieval uncertainty

The uncertainty characterizing VESPA-22 retrieved profiles can be divided in three major contributions: 1) the uncertainty due to the linear approximation used in Eq. (26); 2) the systematic uncertainty due to the uncertainties of the various parameters used in the spectra calibration and pre-processing; and 3) the retrieval uncertainty. One additional error source is the limited vertical resolution inherent to concentration vertical profiles obtained by means of this ground-based observing technique. This leads to solution profiles that can be considered a smoothed version of the real atmospheric concentration profiles. In discussing the Optimal Estimation method, Rodgers (2000) suggests that this error, called "smoothing error", should be estimated only if accurate knowledge of the variability of the atmospheric fine structure is available. This approach is used here and the smoothing error is not included in the error estimate.

The first contribution can be evaluated observing the difference $\Delta y_{lin}$ between the fit spectrum without the addition of the second order polynomial, $y_{fit}^*$, and the spectrum obtained using ARTS to calculate the emission expected from the retrieved profile $x$. In the calculation, ARTS does not perform a linear approximation to the general function $f$ described in Eq. (25):

$$\Delta y_{lin} = y_{fit}^* - f\left(x\right) .$$  (38)

The uncertainty $\Delta x_{lin}$ that $\Delta y_{lin}$ causes on the retrieved profile can be calculated with

$$\Delta x_{lin} = G\Delta y_{lin}$$  (39)

and it has a negligible contribution to the total uncertainty (with a maximum of 0.1 % at 70 km altitude).

In order to evaluate the second contribution listed above, the effects on the retrieved profile due to the variation of each single parameter used in the measurements calibration and pre-processing was investigated. The difference between the profile retrieved using the "correct" value of a specific parameter and the retrieval obtained by changing such value by the estimated relative uncertainty of the parameter is considered to be the contribution $\sigma_i$ of this parameter to the total calibration and pre-processing uncertainty. The total uncertainty from these sources is defined systematic uncertainty. Table 3 summarizes the uncertainties of the various parameters involved in the calibration and pre-processing of VESPA-22 spectra; when the uncertainty is a function of altitude the minimum and maximum values of the uncertainty are reported.

The total systematic uncertainty $\sigma_{cal}$ is therefore given by:

$$\sigma_{cal} = \sqrt{\sum \sigma_i^2}$$  (40)

In Figure 9, the relative contributions of the various parameters used in the calibration and pre-processing are shown.



**Table 3: Uncertainties of the various parameters used in the calibration process. When the uncertainty is a function of altitude the minimum and maximum values of the uncertainty are reported.**

| Parameter | Uncertainty (relative or absolute) |
|---|---|
| Signal angle $\theta$ | $\pm 0.1°$ |
| Noise Diode brightness temperature $T_{nd}$ | $\pm 1.3\%$ |
| Atmospheric opacity $\tau$ | $\pm 5\%$ |
| Air Temperature profile | $[\pm 2.1 \ \pm 5.0]$ K |
| Geopotential height | $[\pm 30 \ \pm 110]$ m |
| Delrin opacity $\tau_d$ | $\pm 10\%$ |
| Spectroscopic parameters | $[\pm 1\% \ \pm 10\%]$ |

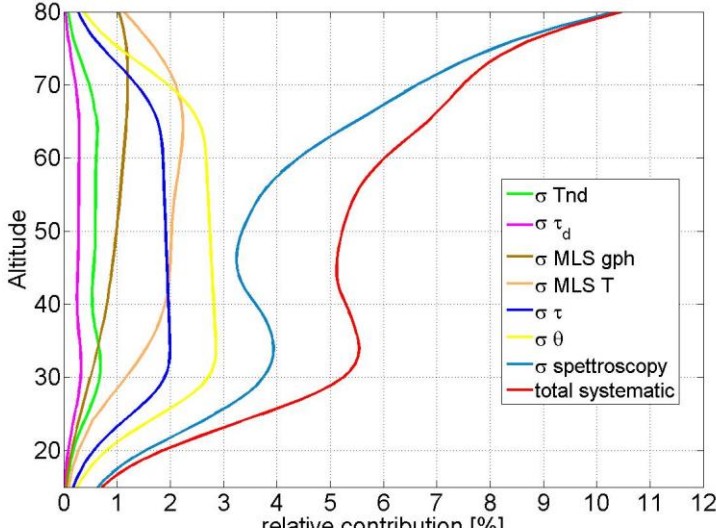

**Figure 9: Relative contributions to systematic uncertainty (red curve). The contributions are: the signal beam angle (yellow), the noise diode temperature (green), the opacity (blue), the MLS meteorological profile (orange and brown), the compensating sheet (Delrin®) opacity (magenta), the spectroscopic parameters (cyan).**

The yellow line shows the $\sigma_i$ contribution due to the uncertainty on the signal beam angle. In order to minimize this contribution, the choked antenna and the parabolic mirror are aligned using a He-Ne laser. During the period from April to October, the VESPA-22 pointing offset can be verified by scanning rapidly at around the sun elevation angle and comparing the measured position of the center of the sun against the known ephemerides. For the measurements discussed here an offset $\Delta\theta = 0.2° \pm 0.1°$ was estimated.





The green solid line shows the potential relative error on the water vapor mixing ratio vertical profile due to the uncertainty on the noise diode temperature $T_{nd}$. In order to evaluate the uncertainty of $T_{nd}$, the 0.4% difference between the values obtained by means of the skydip and the LN$_2$ calibrations (see Sect. 4.3 and Figure 4) is considered. On top of this, also the fluctuations of the signal produced by the calibration diode must be taken into account. These can be evaluated by using the

standard deviation of the difference over time between the $T_{nd}$ values of the two noise diodes, measured to be 1.2%. These two sources of error (0.4% and 1.2%) are added in quadrature to obtain the $T_{nd}$ uncertainty of 1.3%.

The blue line shows the contribution due to the uncertainty $\Delta\tau$ on the sky zenith opacity $\tau$. In order to estimate this uncertainty, the uncertainties introduced by sky inhomogeneity and the estimation of the effective tropospheric temperature $T_{atm}$ are considered. The first contribute it is about 2% but can increase depending from the sky condition. It is estimated

using the uncertainty of the slope parameter produced by the fit used in the tipping curve procedure (see the Sect. 4.3). The second contribute can be evaluated observing the daily fluctuations of $T_{atm}$ measured at the meteorological station of Eureka (Northern Canada), Aasiaat and Alert (Southern Greenland and Canada respectively). The natural day to day temperature fluctuations and the lack of tropospheric meteorological data at Thule during long intervals of time led us to estimate an uncertainty on $T_{atm}$ of about 6 K, which makes the total uncertainty $\Delta\tau$ to be 5 %.

The brown and orange lines show the $\sigma_i$'s due to the temperature and geopotential height uncertainties in the meteorological profile used in the forward calculation. The uncertainties on these parameters are obtained from the MLS data quality and description document (Livesey et al., 2015).

The magenta line shows the contribution of the uncertainty on the compensating sheet opacity, $\Delta\tau_d$. This value is measured observing the effect of the Delrin sheet on the incoming signal from the zenith direction. During regular measurement

conditions, signal and reference beams are balanced with the signal angle at $\theta$ :

$$\overline{T}_S\left(\theta\right)=\overline{T}_R \ . \tag{41}$$

where $\overline{T}_S$ and $\overline{T}_R$ are the mean values of signal and reference beams respectively. The compensating sheet modifies the reference beam according to:

$$T_R = T_{R\_nod}e^{-\tau_d} + T_d\left(1-e^{-\tau_d}\right), \tag{42}$$

where $T_d$ is the sheet physical temperature and $T_{R\_nod}$ is the intensity of the reference beam without the sheet. In order to estimate $\tau_d$, during normal data taking operations the signal angle is locked to its balanced position and the Delrin sheet is removed from the reference beam. Using Eq. (42) and (41), $\tau_d$ can be calculated as:

$$\tau_d = -\ln\left(\frac{T_d-\overline{T}_S}{T_d-\overline{T}_{R\_nod}}\right), \tag{43}$$

Where $\overline{T}_{R\_nod}$ is the mean intensity of the reference beam without the sheet. The brightness temperature of $T_s$ and $T_{R\_nod}$

are calculated according to Eq. (15); the calibration parameters $\alpha$ and $T_{rec}$, are obtained from a LN$_2$ calibration.





In order to carry out the mentioned procedure, qualified personnel must be at the observatory and therefore $\tau_d$ was measured only in July and November, 2016, and February 2017 (see Table 4). Were performed seven measurements of the grey body opacity, two during July, three during November and two during February. The half of the difference between the minimum and maximum values obtained during the same period is used as measurement uncertainty. The measurements mean values

changes with time. This is caused by the degassing typical behavior of plastic sheets, i.e., the property to absorb/release water vapor molecules from/to the environment, which depends on atmospheric humidity. During wintertime the air is drier and the compensating sheets release some water vapor and lower their own opacity.

**Table 4: The mean value and standard deviation of the measured delrin opacity during different periods**

| Thickness | July 2016 | November 2016 | February 2017 |
|---|---|---|---|
| 9 mm | $0.159 \pm 0.004$ | $0.128 \pm 0.003$ | $0.123 \pm 0.003$ |
| 5 mm | $0.088 \pm 0.001$ | $0.072 \pm 0.001$ | $0.070 \pm 0.001$ |

VESPA-22 spectral data are calibrated using a linear interpolation between the $\tau_d$ values measured over time. Although the uncertainty of the measurements is about 2%, the value of the Delrin sheet opacity varies through time and the use of a linear interpolation suggests caution. $\Delta \tau_d$ has therefore been conservatively estimated to be 10%.

The cyan line in Figure 9 shows the contribution due to uncertainties in spectroscopic parameters. This spectroscopic

contribution has the largest impact on the systematic uncertainty. Following Tschanz et al. (2013), the emission line intensity and the pressure broadening parameter were assigned an uncertainty of $8.7 \ 10^{-22} \ m^2$ Hz and of 1014 Hz Pa$^{-1}$, respectively.

Figure 10 shows the results of the retrieval of the VESPA-22 data from October 2016 to May 2017 analyzed with the spectroscopic model eventually adopted for the analysis of VESPA-22 spectra, hereafter reference model, and the same data analyzed with other spectroscopic models. As stated in Sect. 5.1, the reference model line intensity is taken from the JPL

2012 catalogue, modified by adding the hyperfine splitting of the $H_2O$ line (Tschantz et al., 2013) while the pressure broadening parameters are taken from the work of Liebe (1989). Parameters coming from HITRAN (Rothman et al., 2012) and JPL catalogues (Pickett et al., 1998) from different years were compared with one another, as described in the work of Haefele et al. (2009). The JPL catalogue does not include the pressure broadening parameters, so JPL models were integrated with information coming from the work of Liebe (1989) or Cazzoli et al. (2007).

Figure 10, panel a, shows the mean profiles retrieved using the different models. The blue line represents the mean profile obtained using the reference model. Figure 10, panels b and c, shows the mean absolute and relative difference between the retrieved profiles obtained using other spectroscopic models and the reference model. It can be noticed that the water vapor mixing ratio retrieved profile strongly depends on the spectroscopic model of choice, with a relative difference between profiles obtained using different models reaching a maximum of 10% at the top of the sensitivity range. The correlation

coefficient (Figure 10, panel d) of the datasets obtained with the different models respect to the reference model dataset  is





above 0.9 in the sensitivity range. This strong correlation indicates that the time series variation of the water vapor profile measured by VESPA-22 is quite independent from the model used.

**Figure 10: (a) mean retrieved water vapor profiles obtained using different spectroscopic models (the blue line is the model eventually adopted for the analysis of VESPA-22 spectra, called reference model) and vertical profiles of (b) and (c) the mean absolute and relative difference and (d) the correlation coefficient between VESPA-22 profiles obtained using different spectroscopic models and the reference model. The data represented here range from 4 October, 2016, to 22 May, 2017. The retrieval sensitivity range is marked by the dashed horizontal green lines. The spectroscopic parameters are taken from different versions of the JPL catalogue (versions of years 1985, 2001 and 2012) with the pressure broadening parameter taken from the works of Liebe (1989) or Cazzoli et al. (2007) or from the HITRAN catalogue (version of years 2004 and 2012).**





Figure 11 shows the uncertainty calculated for the retrieval shown in Figure 7 of the 23 December 2016. The overall systematic uncertainty is shown with a red curve. The retrieval uncertainty can be evaluated using the S uncertainty matrix obtained as (Rodger, 2000)

$$S = (K^T S_e^{(-1)} K + S_a^{(-1)})^{(-1)} . \tag{44}$$

The square root of the diagonal elements of S represents the uncertainty of the retrieved profile at different altitudes and is indicated in Figure 11 with a blue line. The retrieval uncertainty depends on the level of noise of the spectrum. Finally, the total uncertainty is obtained as

$$\sigma_{tot} = \sqrt{\sigma_{cal}^2 + \sigma_{ret}^2} \tag{45}$$

and is represented with a purple line in Figure 11.

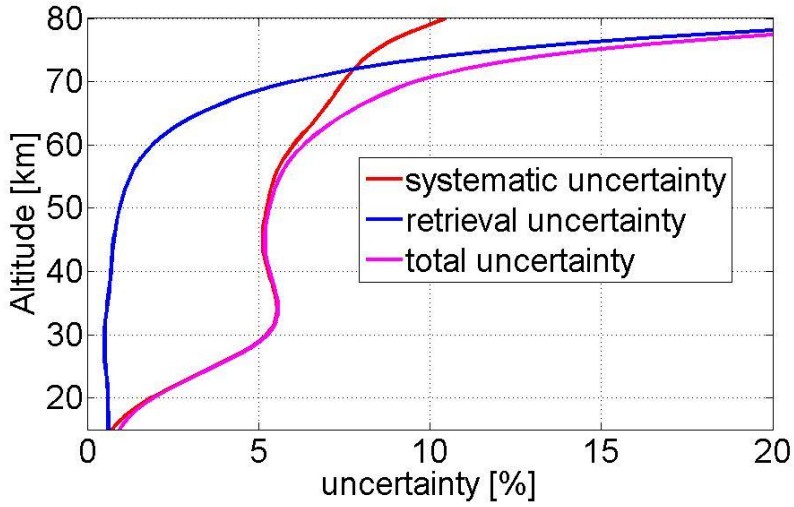

**Figure 11: Vertical profiles of the systematic uncertainty (red), the retrieval uncertainty (blue), and the total uncertainty (magenta) of VESPA-22 water vapor mixing ratio vertical profiles obtained inverting a 24-hour integration spectrum collected on 23 December 2016.**

## 7 VESPA-22 and Aura/MLS water vapor profiles intercomparison

VESPA-22 water vapor vertical retrievals obtained from 24-hour integration spectra (from 00:00 to 23:59) have been compared with version 4.2 of AURA/MLS water vapor vertical profiles. The MLS profiles used for this intercomparison are daily mean profiles obtained averaging all MLS profiles collected within a radius of 300 km centered around VESPA-22 observation point (74.8° N, 73.5° W, see Sect. 5.1). A measurement of the horizontal resolution of VESPA-22 is the



horizontal area highlighted by the half power full beam angle; for an observation angle of 15° at a height of 60 km it is an ellipse with axes of about 55 x 14 km.

The intercomparison is carried out using data from 15 Jul 2016 to 21 May 2017. The spectra collected during July, August and September are less continuous and noisier due to testing of the equipment, poor weather conditions, and snow covering

the zenith observing window (in November 2016 a more powerful fan was installed outside the windows to minimize snow deposition). Additionally, there are no measurements carried out by VESPA-22 between 4 and 22 November, 2016, and between 12 and 16 February, 2017, due to poor weather conditions and snow covering the reference beam window, and from 11 to 16 December, 2016, due to technical issues. During the second part of March, April and May 2017, the spatial distance between MLS and VESPA-22 profiles increases, on average; the MLS profiles are composed averaging less data and there

are several days in this period with no concurrent MLS data. A few isolated days in which large sky inhomogeneities do not allow the correct balance of signal and reference beams have also been removed from the intercomparison. It is worth recalling that the signal to noise ratio of VESPA-22 spectra, and therefore the quality of the retrievals, depends on the sky opacity and, consequently, on the season, being noticeably better during winter and poorer in summer. This is particularly true for microwave instruments installed in the Polar regions, where the seasonal fluctuations in tropospheric water vapor

column content are significant.

Table 5 summarizes the characteristics of the Aura/MLS water vapor version 4.2 retrievals (Livesey et al., 2015).

**Table 5: The main specifications of Aura/MLS water vapor mixing ratio profiles version 4.2 (Livesey et al., 2015)**

| Pressure [hPa] | Altitude [km] | Resolution V x H  [km] | Single profile precision [%] | Accuracy [%] |
|---|---|---|---|---|
| 0.002 | 86 | 10.3 x 350 | 152 | 34 |
| 0.01 | 76 | 8.8 x 725 | 55 | 11 |
| 0.046 | 66 | 7.4 x 540 | 35 | 8 |
| 0.21 | 55 | 3.6 x 670 | 19 | 7 |
| 1.0 | 44 | 2.5 x 400 | 6 | 4 |
| 4.6 | 35 | 3.4 x 350 | 4 | 7 |
| 22 | 26 | 3.2 x 265 | 5 | 7 |
| 68 | 18 | 3.1 x 190 | 5 | 6 |

In order to compare the two datasets, MLS vertical profiles are convolved with VESPA-22 averaging kernels in order to match the resolution of VESPA-22 profiles according to:

$$x_{MLS} = x_a + A\left( \tilde{x}_{MLS} - x_a \right) \tag{46}$$

where $\tilde{x}_{MLS}$ is the raw (high resolution) MLS water vapor vertical profile, $x_a$ is the apriori profile, A is Averaging Kernel Matrix and $x_{MLS}$ is the convolved MLS profile.





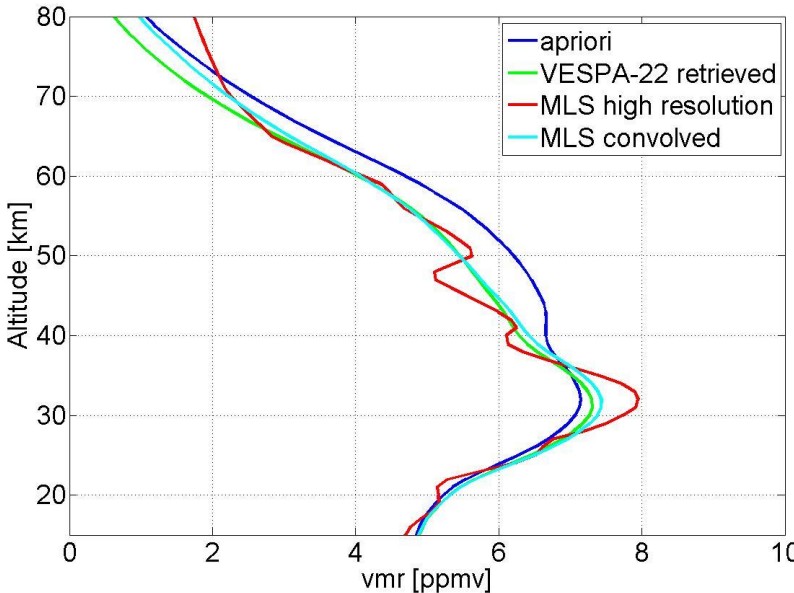

**Figure 12: the MLS profile measured on 23 December 2016 with high vertical resolution (red line) compared with the convolved profile obtained from Eq. (46) and with the VESPA-22 retrieved profile (green line) and the apriori profile (blue line).**

Figure 12 shows the MLS measured profile on 23 December 2016 (red line) convolved using Eq. (46) (cyan line), together

with the apriori profile (blue line) and the VESPA-22 retrieved profile (green line). The MLS convolved profile tends to the apriori profile below 25 km and above 72 km, where the retrieval sensitivity drops.

Figure 13, panel a, shows the mean VESPA-22 retrieved profile (in blue) and the mean MLS convolved profile (in red), with their standard deviations indicated with dashed lines, whereas panel b shows the mean sensitivity of VESPA-22 retrieved profiles with its standard deviation (solid and dashed lines respectively), which is larger than 0.8 from 25 to 72 km altitude.

This interval can vary from day to day depending from the noise level of the 24-hour integrated spectrum. Panels c and d display the relative and absolute differences of VESPA-22 water vapor mixing ratio mean vertical profile with respect to the MLS mean convolved profile (red line) and with respect to the MLS high resolution mean vertical profile (i.e., not convolved; blue line). The largest relative and absolute differences between the two datasets occurs at 72 km, the upper limit of the VESPA-22 sensitivity range, and are about -6% and -0.2 ppmv, respectively, with Aura/MLS convolved mean profile

being larger than VESPA-22 mean retrieval.



**Figure 13:** (a) VESPA-22 averaged water vapor mixing ratio vertical profile with its standard deviation (blue line and blue dotted lines) and the mean MLS convolved profile with its standard deviation (red line and red dotted lines); (b) the mean retrieval sensitivity profile; (c) relative and (d) absolute differences vertical profiles between VESPA-22 and MLS convolved mean profiles (red line), and between VESPA-22 and MLS high resolution (not convolved) mean profiles (blue line); (e) vertical profiles of the correlation coefficient between VESPA-22 and MLS convolved profiles (red line), and between VESPA-22 and MLS high resolution profiles (blue line); (f) vertical profile of the mean full width at half maximum of VESPA-22 averaging kernels. The data used for the intercomparison range from 15 July 2016 to 21 May 2017.





The mean difference between the two datasets at the extremes of the sensitivity range could be reduced using an apriori for VESPA-22 retrievals that is closer to the real atmospheric state, as it would be for example a monthly average of MLS mean profiles collected during the same month of the measurement. However, as already discussed in Sect. 5.1, a climatological

apriori, i.e., the monthly average of three years (2014-2016) of MLS data has been chosen instead, as this approach allows the identification and study of potential interannual water vapor variations. In Figure 13, panel e, the vertical profiles of the correlation coefficient between VESPA-22 and MLS convolved data (red), and between VESPA-22 and MLS not convolved data (blue) are shown. The former correlation is about 0.9 or higher over the entire sensitivity range while the latter is between 0.7 and 0.94. The correlation between VESPA-22 and MLS high resolution water vapor profiles is useful to

illustrate how the two completely independent datasets correlate over the extended altitude range from 10 to 80 km.

Figure 13, panel f, shows the vertical profile of mean values of the full width at half maximum (FWHM) of VESPA-22 averaging kernels calculated over the comparison period with its standard deviation (blue dashed lines). The FWHM of the averaging kernels is a measure of the retrieval vertical resolution (Rodgers, 2000). The single profile vertical resolution can vary depending from the level of noise affecting the measured spectrum.

The correlation between the water vapor mixing ratio profiles of VESPA-22 and Aura/MLS can be evaluated also by looking at Figure 14, where MLS values at three different altitude levels (high resolution in blue and convolved in red) are displayed as a function of the VESPA-22 values at the same altitudes, with the green line representing the linear regression of the MLS convolved vs. VESPA-22 measurements and the orange lines the ideal case of perfect agreement. Table 6 displays the parameters of the three linear regressions.

**Table 6: Values obtained from the linear regressions between MLS convolved and VESPA-22 measurements at the indicated altitudes.**

| $y = ax + b$ | $a$ | $b$ [ppmv] |
|---|---|---|
| 35 km | 0.92 | 0.52 |
| 50 km | 0.91 | 0.69 |
| 65 km | 1.01 | 0.09 |

Figure 15 shows the time series of water vapor mixing ratio values measured at different altitudes by means of VESPA-22

(blue dots) and obtained by convolving MLS original vertical profiles (red dots). The time series at 50 km and 65 km altitude show some oscillations in the MLS convolved measurements after the 13th of March, possibly due to the increased distance in this period between VESPA-22 and MLS observed air masses and the less available data to compute the MLS daily averaged profiles. In the same period the increasing sky opacity and poor weather produced nosier VESPA-22 measurement, decreasing the retrieval sensitivity.




In order to provide a more complete, albeit less quantitative, overview of the VESPA-22 and convolved MLS time series, Figure 16 shows contour maps of VESPA-22 (in color) and MLS convolved (black line) water vapor profiles. The data collected by both instruments revealed an absolute maximum in August 2016 at 50 km of height of about 8.3 ppmv. During fall and winter the maximum of the water vapor mixing ratio profile lowers its altitude to about 30 km in January 2017, due

5   to the air subsidence inside the polar vortex. A steep gradient in water vapor mixing ratio is observed between 25 and 40 km altitude in the second half of January, due to the polar vortex that shifts away from Thule for about two weeks. Both instruments also observe the return of the water vapor mixing ratio to pre-winter values in mid-April, possibly indicating the occurrence of the vortex final warming, with the re-establishment of a maximum in the mixing ratio profile at about 50 km altitude.

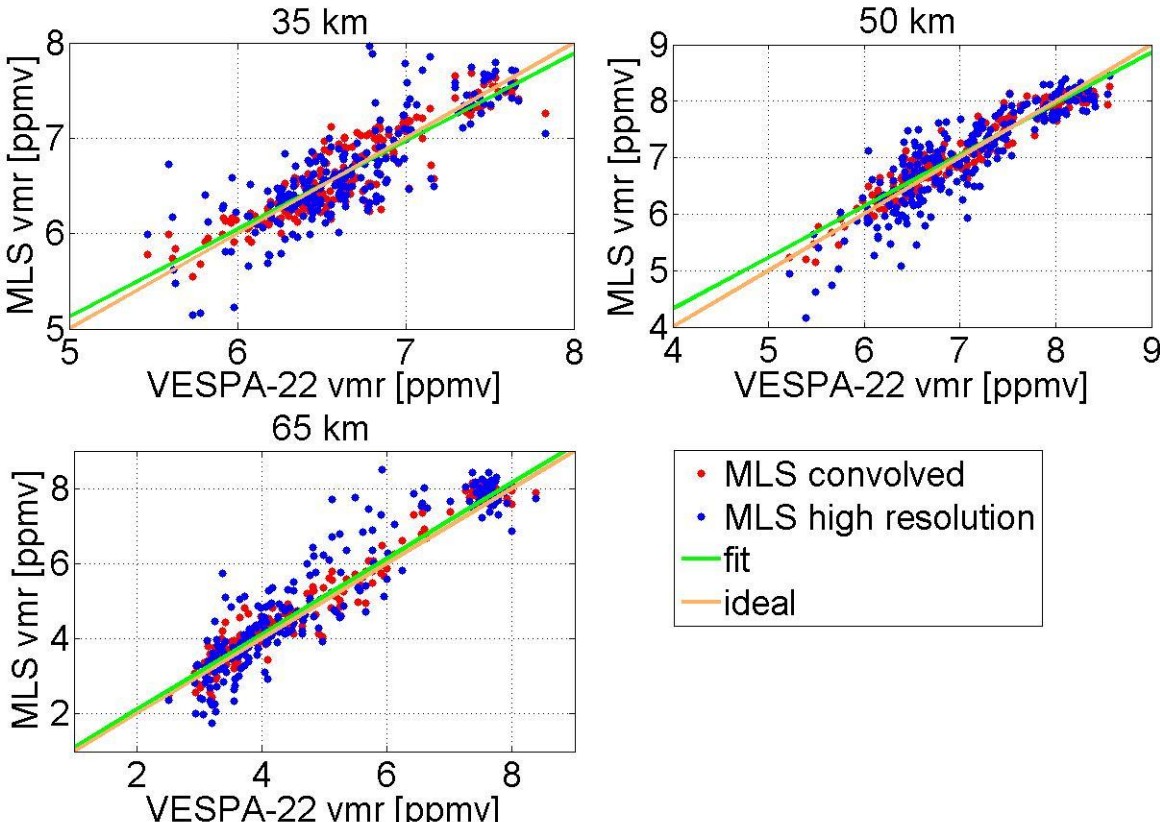

**Figure 14: High resolution (blue dots) and convolved (red dots) MLS water vapor mixing ratio values at three different altitudes plotted as a function of the corresponding VESPA-22 measurements obtained during the intercomparison period. The green line shows the linear regression between MLS convolved and VESPA-22 measurements while the orange line represents the ideal case**
15   **of perfect agreement. The data used for the intercomparison range from 15 July 2016 to 21 May 2017.**





Given the large variability that stratospheric water vapor mixing ratio profiles can experience in Polar regions, in particular during winter and spring, Figure 17 is meant to illustrate the variations of VESPA-22 averaged mixing ratio profiles over the eleven months of reported measurements, and how the difference between VESPA-22 and MLS profiles varies from month to month. Figure 17, panel a, shows the monthly average profiles collected by VESPA-22, whereas panel b shows their

5   relative difference with the monthly average profiles of convolved MLS. The maximum of mixing ratio in summer can be noticed in panel a, decreasing over time and lowering its altitude from September to January. The relative difference between the two datasets shows a similar shape during fall and winter, with a maximum absolute difference at the top of the sensitivity range from -5% to -10%, depending from the month. During summer the VESPA-22 data are characterized by a bias of about +7% at the lower extreme of the sensitivity range, that slowly disappear in October. During April and May the

10  Vespa-22 and MLS have a better agreement but it is due to the reduced sensitivity of VESPA-22 retrieval.

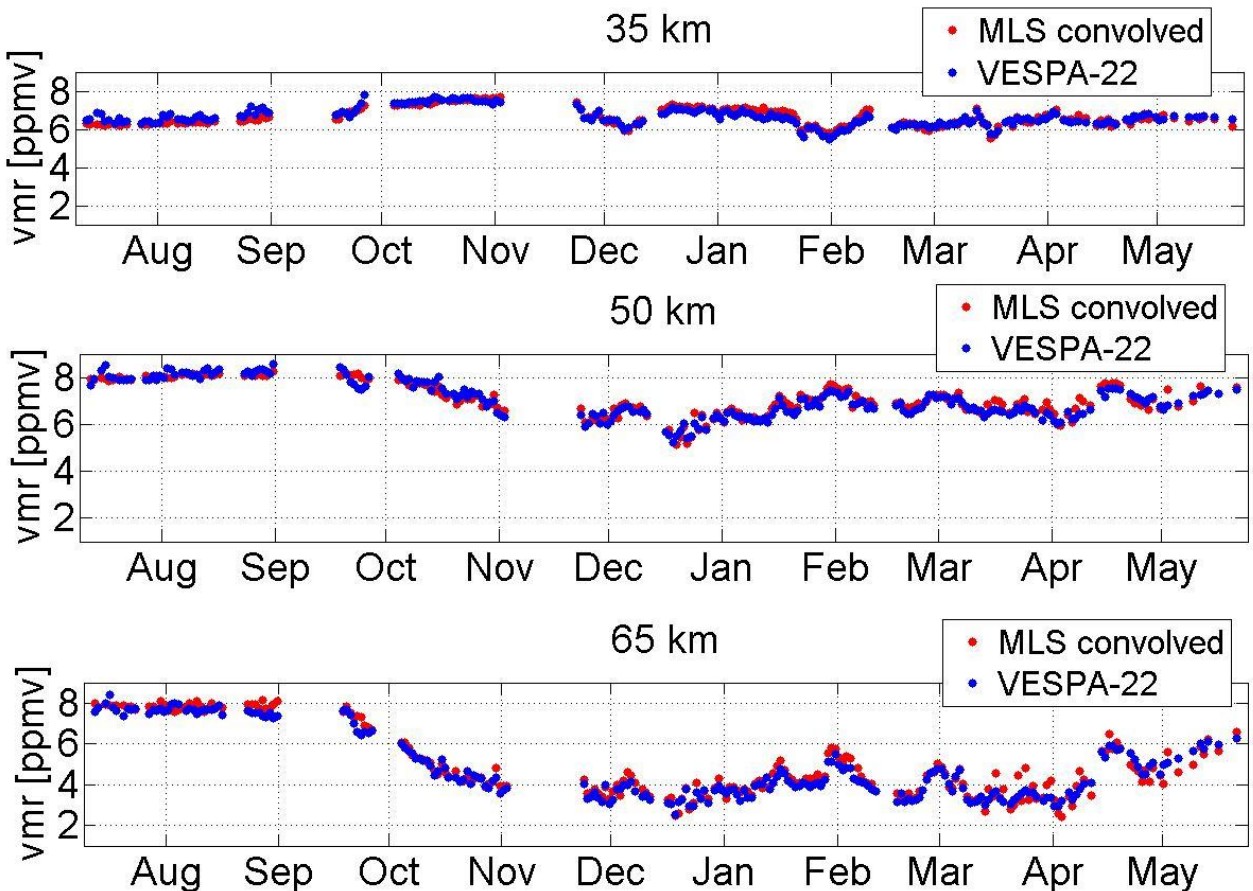

**Figure 15: Time series at different altitudes of water vapor mixing ratio values obtained with VESPA-22 (blue) and by convolving MLS vertical profiles (red).**





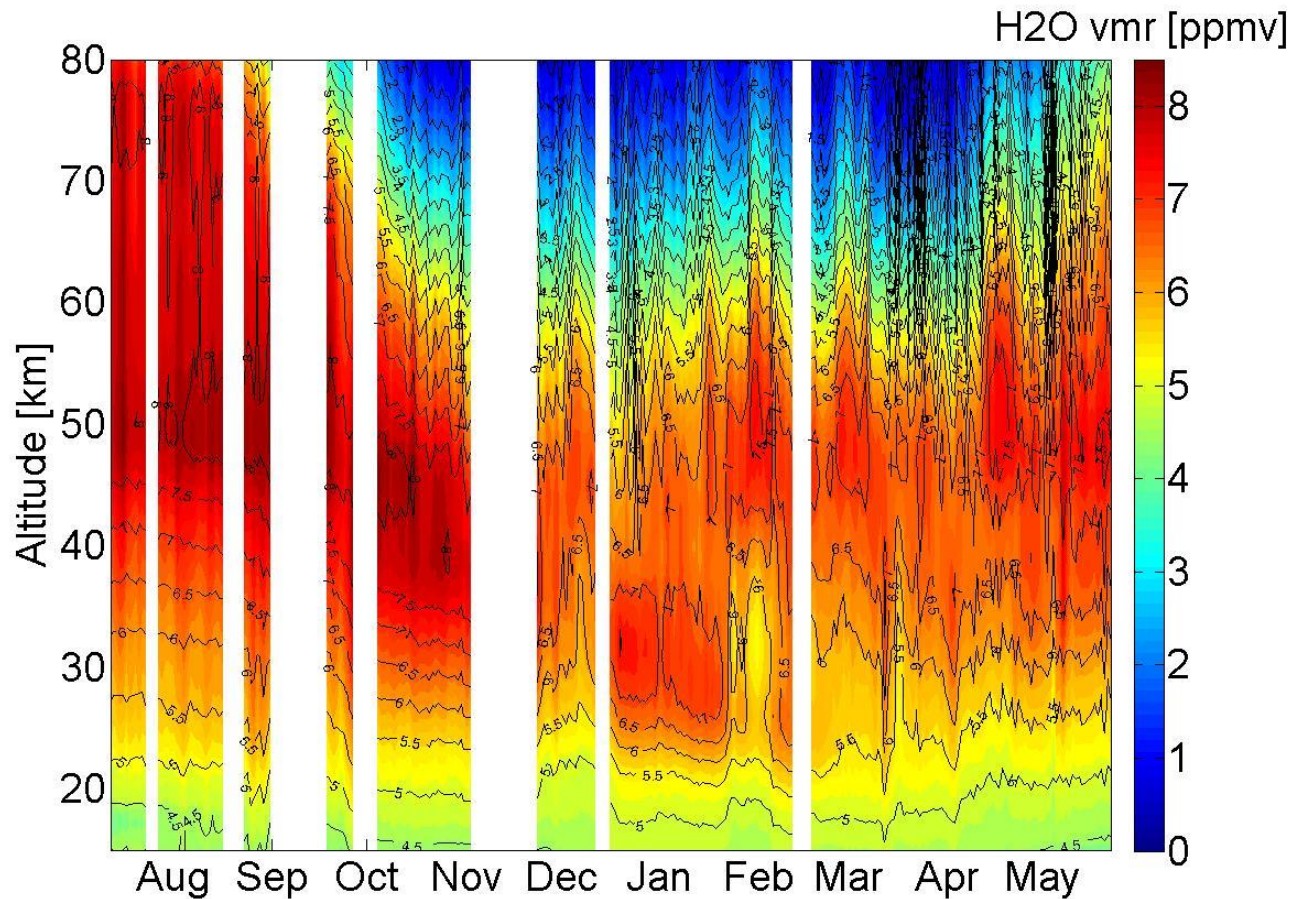

**Figure 16: a map showing the VESPA-22 retrieved profiles (colored map) compared to MLS convolved profiles (black lines). The blank areas characterize the interruption period longer than three days.**




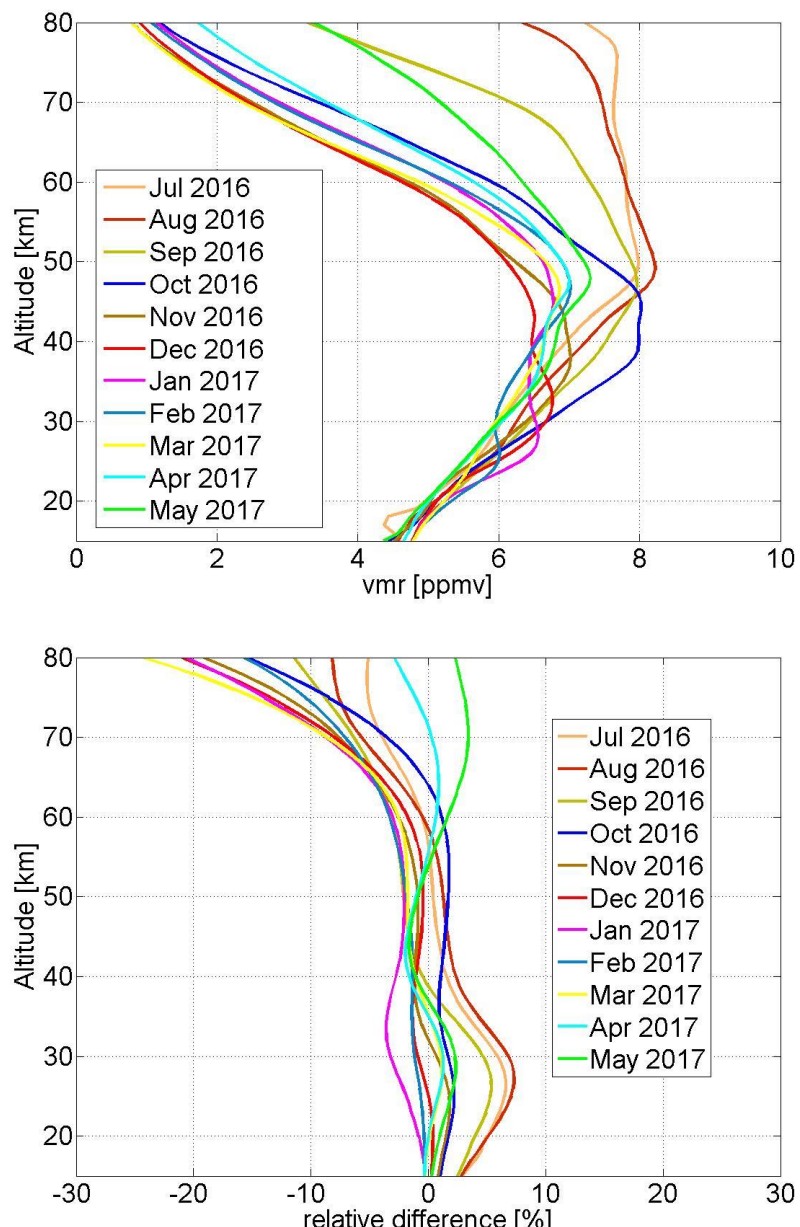

**Figure 17: (a) Monthly averages of the VESPA-22 water vapor mixing ratio profiles, and (b) monthly averages of the difference between VESPA-22 and convolved MLS dataset.**



## Summary

VESPA-22 was installed at the Thule High Arctic Atmospheric Observatory (THAAO) located at Thule Air Base
(Greenland) in July 2016 for long-term observations of Polar middle atmospheric water vapor. The instrument is
characterized by a full beam at half power encompassing 3.5°, granting the observation of the signal beam at angles as low
as about 12° above the horizon. The instrument is installed indoor in a wooden annex to the main laboratory and observes the
sky emission through observation windows made of 5-cm thick Plastazote LD15 sheets. There are no evident artifacts larger
than 2 mK affecting the measured spectra. VESPA-22 operated automatically with minimum need of maintenance for about
one year, proving the robustness of hardware and acquiring system.

Instrument calibration is regularly performed using two noise diodes. The emission temperature of these elements is
measured during tipping curve calibrations and checked periodically against liquid nitrogen calibrations. The calibrating
noise diode temperature is estimated with an uncertainty of 1.3%, evaluated confronting calibration results obtained by
means of tipping curves and liquid nitrogen, and the standard deviation of the differences between the two noise diodes time
series.

VESPA-22 retrieval algorithm is based on the optimal estimation technique; the retrieved profiles from 24-hour integration
spectra have a sensitivity larger than 0.8 from about 25 to 72 km of altitude and a vertical resolution from about 12 to 23 km.
The forward model is provided by the ARTS software (Eriksson et al., 2013) and it simulates the VESPA-22 measurement
technique. In order to perform a realistic simulation of the troposphere, the PWV measured by the HATPRO radiometer
operating side by side with VESPA-22 is taken into account in the forward model.

The uncertainty is evaluated as the sum of the contributions of calibration, pre-processing and spectroscopic parameters and
measurement noise. In the sensitivity range of VESPA-22 retrievals, the total uncertainty is about 5-6 % from 26 to 60 km
increasing to about 12% at 72 km.

VESPA-22 and MLS retrieved profiles are compared during a period from July 2016 to May 2017. The VESPA-22 data used
in the comparison are the results of retrievals from 24-hour integration spectra, from 00:00 to 23:59, while MLS data are the
daily mean profiles collected by MLS in a radius of 300 km around VESPA-22 observation point. VESPA-22 and MLS
convolved datasets show a good correlation with a correlation coefficient of about 0.9 or higher. No significant bias between
the two datasets was observed in the altitude range from 25 to 60 km, whereas in the altitude range from 60 to 72 km the
value of the mean difference (VESPA – MLS) increases reaching -6% (-0.2 ppmv) at 72 km.

The results described in this manuscript proved that VESPA-22 is capable of carrying out reliable water vapor stratospheric
measurements during different seasons and weather conditions, even during spring and summer, although the data collected
in these periods are affected by larger noise with respect to other seasons, due to the larger amount of tropospheric water



vapor. VESPA-22 is capable of observing the seasonal variations of the water vapor concentration vertical profile in the stratosphere, as for example the water vapor subsidence occurring inside and at the edge of the polar vortex. The VESPA-22 and MLS convolved retrieved monthly averaged profiles show an agreement within 10%, with the maximum difference measured during winter at the top of the sensitivity range (Figure 17, panel a and b). Furthermore, VESPA-22 retrievals correctly represent the rapid variations that can occur in the stratosphere, as demonstrated by the large water vapor gradients measured in late January/early February and then during April and May (see Figure 16) by both VESPA-22 and Aura/MLS.

*Acknowledgements*. Bob de Zafra, Mike Gomez, and Axel Murk helped with the development of VESPA-22 by suggesting useful solutions to technical problems related to the instrument design. We are indebted to them. Additionally, G. Muscari is very thankful to Nik Kaempfer and Axel Murk for donating to the VESPA-22 project the firmware that the FFTS is currently employing. We are grateful to Stephan Buehler and Patrick Eriksson for helping with the ARTS software, and to the MLS team of the Jet Propulsion Laboratory for providing access to the Aura/MLS data.

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
