# Peer review of "VESPA-22: a ground-based microwave spectrometer for long-term measurements of Polar stratospheric water vapor"

_Atmospheric Measurement Techniques, 2017_

## Referee Comment (RC1) · Anonymous Referee #1 · 14 Sep 2017

This manuscript nicely describes the VESPA-22 instrument for measuring water vapor, and provides important details of the data analysis technique that seem to me to be appropriate for AMT. I do, however, have some significant concerns with several important claims made in the manuscript.

Main concerns:

There is a second order polynomial which is "added to the retrieval", but there is no discussion of the details of this process. If it is fit individually during each retrieval, and is properly included in the error analysis, then it will have a large effect on the lower stratospheric sensitivity. I am therefore somewhat skeptical of the accuracy of the

sensitivities shown in Figure 8 at ∼30km and below. The other choice is to treat this as a systematic term, in which case it affects the systematic error but not the sensitivity. In either case, this can introduce an important uncertainty, and the authors need to discuss exactly what they have done.

In the error analysis, many of the terms have been classified as systematic, when in truth, with the exception of the spectroscopy, almost all of them have a significant random component. As a result, the "retrieval uncertainty" shown in Figure 11, which by implication is the only random component, is, in my estimation, absurdly small. I think it would be acceptable to not specifically label the bulk of the errors as either completely systematic or random, but the present suggestion that the precision is <1% over much of the atmosphere would require a great deal of additional evidence.

The VESPA retrievals are performed using a seasonally varying a priori based upon 3 years of seasonally varying MLS data, hence the statement on page 29 that these are "independent datasets" is not true. Claims of high correlation between the measurements are therefore unsubstantiated. If the authors wish to publish claims related to correlations they should either perform their retrievals with a constant a priori (which would probably still leave them with good correlations), or compare deviations from the a priori for the two datasets (which is a tough test). Alternatively they could drop the whole discussion of correlation, along with Table 6 and Figure 14. Also, as long as the retrievals are done with a varying a priori Figure 15 should include a point indicating the a priori for each month.

Equation (4) – presumably 'x' is known for this transition. What is it?

Page 6 line 20 "negligible". Perhaps "small" would be better here. The numbers are given later in the paragraph, so everything here is okay, but for some applications a 1.5% difference might not be considered "negligible".

The use of Tatm is a bit confusing, as this appears to be the atmospheric temperature for the troposphere, but not for other parts of the atmosphere. Would it be better to

label this as Ttrop?

Figure 4 is very nice. Page 10 line 7 – Where does this "constant in frequency within 1.5%" number come from? The diode data sheets? Or perhaps a calibration measurement?

"Tsup" is a rather odd abbreviation for surface temperature.

Page 11 line 7: "explicit" should be "explicitly give"

Page 15 line 13 – "Selee" should be "Seele"

Page 17 line 8 – missing '('

Page 14 line 8 – How is this second-order polynomial calculated? Is it constant? Is it fit for in the retrieval somehow?

Page 17 - The phrase "Figure 6 a shows a VESPA-22 spectrum integrated for second-degree polynomial 24 hours (blue)" does not make sense to me. What exactly is being done here?

Figure 9 –"specttrocscopy" should be "spectroscopy"

---

## Referee Comment (RC2) · Anonymous Referee #2 · 18 Sep 2017

The discussion paper gives an extensive introduction to the VESPA-22 instrument, its calibration and data analysis, as well as a comparison of measurement results with MLS observations. It fits well in the AMTD journal, but it would benefit from a few clarifications and corrections.

In general, the introduction in section 3 to 5 repeats the basics of radiative transfer, balancing instruments and optimal estimation, which are already described in detail in the cited literature. In my opinion, these sections could be easily shortened without loss of information.

[Figure]

Technical Comments:

The abstract (p.1 line 13) and section 3 describe the VESPA-22 back-end as an FFTS with 500 MHz bandwidth and 31kHz resolution. However, the schematic in Fig. 1(b) shows a system with 1GHz bandwidth. Is this the actual schematic of the instrument? Also P4 line 1 states a 2GS/s sampling rate, which results in a 1 GHz Nyquist bandwidth.

P. 3 line 13: The sentence "the waveguide used by VESPA-22 polarizes the incoming radiation with a gain difference between the two polarization modes of 45 dB" is both incorrect and irrelevant. The rectangular waveguide supports only a single polarization by definition. The stated 45dB may refer to the feed horn, but the actual cross-polar with the offset reflector will be much worse. Bertagnolio 2012 states for the same instrument 35dB and 24dB, respectively. This should not affect the observations, so the sentence could be easily removed.

P. 6 line 28 states that the opacity in Fig 3 was calculated for water vapor profiles measured by AURA/MLS. However, the dominant effect in this figure is due to tropospheric water vapor which is not measured by MLS.

P. 9 line 24 states that a 6 MHz wide interval around the line center is kept at 31kHz resolution, while all other channels are binned with 50 channels. In Fig 6(b) the interval without binning looks much smaller than 6MHz. According to p. 14 line 1 the matrix with the measurement error Se was assumed with constant diagonal elements which are calculated from the residuals after an initial run of the retrieval. However, the central channels without binning will have a higher measurement error as the ones without binning. Please provide also the range of measurement errors which were used in Se for the daily retrievals. The retrieval errors in Fig 11 seem to be very small. Could this be an artifact of an underestimation of the measurement errors in Se due to the binning at the line wings?

Where do the values for the apriori covariance in Fig. 5 come from?

[Figure]

P. 14 line 7: "A second-order polynomial is also added to the retrieval ...". At which stage is it taken into account in the retrieval, and how does it affect the measurement response at the lower altitudes? Is it really a second order polynomial, or just a straight line?

P. 10 line 7 "The noise diode produces a signal that can be considered constant in frequency within 1.5%.". Where does this number come from, and can it be demonstrated in the VESPA22 measurements? Over which bandwidth is it valid? Presumably only 40MHz are used for the retrieval, but according to Eq. 20 the mean $T\_ND$ is calculated for a much wider frequency range.

Also the opacity of the Delrin sheet is assumed to be frequency independent. Is this really the case, or does e.g. the polynomial baseline fit change after changing the thickness of the Delrin sheet?

P. 12 line 6 claims that the noise diode temperatures measured with LN2 and tipping curve agree within 0.4%. However, the fluctuations of the tipping curve results in Fig. 4 seem to be in the order of several Kelvin, which should result I a higher discrepancy. Are these fluctuations measurement errors caused e.g. by an inhomogeneous atmosphere, or doe they represent a real fluctuation of the noise diode ENR caused e.g. by changes of the laboratory temperature? Which value was used in the actual retrievals?

P 11 line 3: "$T\_atm$ is the average temperature obtained from radiosonde data by weighting the tropospheric temperature vertical profile with the water vapor concentration profile." Is this weighting done with the water vapor vmr or with vmr*p ?
* * *
Minor comments on language and typos:

Several occasions: replace "depending from" with "depending on"

The abstract (p.1 line 3) mentions an integration time in the order of hours, but the paper discusses only the results of daily mean values.

P. 2 line 17: The statement "Positive trends were observed during the last two decades" is followed by citations from 1999-2001. This sentence should be reworded or backed by more recent citations.

P. 3 line 10: instead of "full beam at half power" the term "full width at half maximum" (FWHM) would be more common and is also used in the cited Bertagnoli 2012 paper

P. 3 line 19: "Two noise diodes are inserted in the IF chain" should read "RF chain". Preferably "IF" and "RF" should be explained at first instance.

P. 4 line 19 mentions the brand name "eccofoam", but earlier the window was identified as LD-15 which is a different material.

P. 9 line 3: The statement "The 'zero' signal is measured and subtracted to every acquired spectrum..." is misleading and could be understood that V0 is measured with every spectrum. Please clarify when and how often V0 is measured and whether it has a significant effect. Presumably it does not contribute at all to the result since the calibration with Eq. 13 and 16 uses always differences between two raw spectra.

P. 9 line 5: "..and subtracts the counts number from these two sources" should probably read "...numbers...".

P. 13 line 16 states that the central 400MHz are used in the retrieval. Since the Fig 68a) shows only 40 MHz this is most likely a typo.

P. 17 line 6+8: There seems to be a copy and paste error in the sentences "... integrated for second-degree polynomial 24 hours . . ." and "The cyan line is the retrieved by the inversion..."

P. 22 line 20 and following: These paragraphs repeat the principles of the Delrin balancing, Eq 42 is very similar to Eq.8. Also the details of the sheets and how their opacity is determined are presented. In my opinion this would fit better to section 4.1 where the instrument and calibration are discussed, than into this section 6 "retrieval uncertainty".

P23 line 30: "...of the datasets obtained with the different models [. . .] respect to the reference model dataset..." Apparently a missing "with" in [. . .], otherwise the sentence does not make sense to me.

Fig. 10: The labels (c) and (d) are missing in the lower two subplots. Fig. 17: The labels (a) and (b) are missing in the two subplots.

---

## Author Comment (AC1) · 21 Nov 2017

**Replies to Reviewer 1**

We thank both reviewers for their knowledgeable and valuable comments. Our efforts in addressing them, together with the reviewers' suggestions, led to a revised manuscript that represents a great improvement with respect to the original version. In what follows, reviewer's comments are in black and authors' replies in red.

There is a second order polynomial which is "added to the retrieval", but there is no discussion of the details of this process. If it is fit individually during each retrieval, and is properly included in the error analysis, then it will have a large effect on the lower stratospheric sensitivity. I am therefore somewhat skeptical of the accuracy of the sensitivities shown in Figure 8 at _30km and below. The other choice is to treat this as a systematic term, in which case it affects the systematic error but not the sensitivity. In either case, this can introduce an important uncertainty, and the authors need to discuss exactly what they have done.

We added to the manuscript a discussion on how the retrieval process includes a second order polynomial. This is done by fitting it in each retrieval, individually. It is not a systematic term. It does have a large effect on the sensitivity below 30 km but this is already taken into account in the sensitivity profile of figure 8 (original manuscript, now figure 9). In the figure below, we show the difference in sensitivity between employing the second order polynomial or only a first order one.

[Figure]

In order to address this issue raised by the reviewer, we estimated the uncertainty in the retrieved profile due to the use of the second order polynomial. The most rigorous way of doing it, in our opinion, is to take the uncertainty associated to the second order coefficient ($\Delta a_2$) in the retrieval process (say for example 20%, e.g., $a_2 = (-5 \pm 1) \times 10^{-3}$) and then perform two retrievals for the same spectrum with fixed values of the second order coefficient equal to $a_2 \pm \Delta a_2$ (in our example they would be $a_2 = -4 \times 10^{-3}$ and $a_2 = -6 \times 10^{-3}$). The resulting two vertical profiles would then provide an estimate for the uncertainty of the regular retrieved profile

associated with the uncertainty in the second order coefficient. We found that the average uncertainty of the second order coefficient calculated by the optimal estimation routine over the entire dataset is 6%, with few retrievals showing more than 20%. We therefore decided to employ a fixed maximum uncertainty on the coefficient of 20% for the whole data set, rejecting from the data set those few retrievals (less than 5% of the total) that had an uncertainty larger than 20% in the determination of $a_2$. Displayed below is the updated figure related to the error analysis of the spectrum observed on 23 Dec, 2016, where we added the polynomial uncertainty to the other 2 sources. As expected the polynomial uncertainty has an impact mostly in the lower part of the retrieval.

[Figure]

On top of it, in order to demonstrate the good quality of VESPA retrievals down to 25 km altitude, we changed the apriori used for the retrievals, which is now a fixed climatological profile up to about 48 km and a seasonal profile in the mesosphere (see also our reply to this specific issue raised by the reviewer in the following), and we introduced in the manuscript the correlation between VESPA profiles and MLS smoothed profiles. These are the original high resolution MLS profiles smoothed in the vertical with a running average of 10 km. This was done in order to decouple MLS profiles from the apriori and the averaging kernels used in VESPA retrievals (which was a short-coming of the correlation between VESPA and MLS convolved profiles shown in the originally submitted manuscript) and yet make the MLS vertical resolution somewhat similar to that characterizing VESPA profiles.

Figures down below show the relative and absolute average differences between VESPA and MLS smoothed (blue), and between VESPA and MLS convolved (red),

the correlation coefficient between VESPA and MLS smoothed profiles (blue),

and the time series at 25 km of VESPA22, MLS convolved and MLS smoothed.

[Figure]

All the figures above demonstrate in our opinion that VESPA22 retrievals are scientifically valuable in the sensitivity range indicated in the manuscript. Please note, however, that we also specified in the revised manuscript that the sensitivity range changes with seasons (see fig. 18 in the revised manuscript) and in summer it is approximately between 30 and 65 km. See figure below.

[Figure]

What was changed in the revised manuscript:

- Added discussion in Section 4 on how the second order polynomial is treated in the retrieval process;
- Modified figure 12 with the updated error analysis which includes the polynomial uncertainty;
- Modified figure 9 according to the new error analysis;
- Inserted the MLS smoothed data set, with its correlation with Vespa-22 profiles;

- Introduced the time series of the sensitivity interval, i.e., how the interval of accepted sensitivity changes through time. Added this time series in figure 18;
- Added two altitude levels in former figure 15 (figure 17 in the revised manuscript) to prove the good quality of VESPA-22 retrievals at the lower (25 km) and upper (75 km) limits of the sensitivity interval.

In the error analysis, many of the terms have been classified as systematic, when in truth, with the exception of the spectroscopy, almost all of them have a significant random component. As a result, the "retrieval uncertainty" shown in Figure 11, which by implication is the only random component, is, in my estimation, absurdly small. I think it would be acceptable to not specifically label the bulk of the errors as either completely systematic or random, but the present suggestion that the precision is <1% over much of the atmosphere would require a great deal of additional evidence.

We agree with the reviewer and changed the terminology accordingly. The uncertainty due to spectroscopic and calibration parameters is now indicated with "calibration uncertainty" in the revised manuscript.

The VESPA retrievals are performed using a seasonally varying a priori based upon 3 years of seasonally varying MLS data, hence the statement on page 29 that these are "independent datasets" is not true. Claims of high correlation between the measurements are therefore unsubstantiated. If the authors wish to publish claims related to correlations they should either perform their retrievals with a constant a priori (which would probably still leave them with good correlations), or compare deviations from the a priori for the two datasets (which is a tough test). Alternatively they could drop the whole discussion of correlation, along with Table 6 and Figure 14. Also, as long as the retrievals are done with a varying a priori Figure 15 should include a point indicating the a priori for each month.

We followed the reviewer's suggestion and changed the whole analysis by using only two apriori profiles during 10 out of the 12 months of the year, identical below 48 km and diverging above, due to the large difference in Polar regions between summer and winter water vapor mesospheric profiles. We added figure 7 in the revised manuscript (and below) showing the two apriori profiles. The summer apriori (red line) is used during the period from June 1 to August 31, while the winter apriori (blue line) is used during the period from October 1 to April 31. During the month of May and September, there is a transition period in which we used a linear daily interpolation from one apriori to the other to provide continuity in the retrieved profiles timeseries.

We also added the time series of the apriori mixing ratio values in figure 17 (former figure 15) as suggested by the reviewer.

[Figure]

Page 10 line 7 – Where does this "constant in frequency within 1.5%" number come from? The diode data sheets? Or perhaps a calibration measurement?

The indicated 1.5% is half of the maximum difference between brightness temperature values over the spectral passband (see figure below). The noise diodes emission spectra are measured only during the LN$_2$ calibration. In the revised manuscript we rephrased the "constant in frequency" sentence with "The noise diode produces a signal that is measured to be quite stable in frequency. In fact, single-channel $T_{nd}$ values are always within 1.5% of the spectral mean of the diode temperature brightness $T_{nd}$."

[Figure]

Equation (4) – presumably 'x' is known for this transition. What is it?

We added a sentence to direct readers to Table 2 where all the spectroscopic parameters used are listed

Concerning all the comments below, we agree with the reviewer and changed the manuscript accordingly.

Page 6 line 20 "negligible". Perhaps "small" would be better here. The numbers are given later in the paragraph, so everything here is okay, but for some applications a 1.5% difference might not be considered "negligible".

The use of Tatm is a bit confusing, as this appears to be the atmospheric temperature for the troposphere, but not for other parts of the atmosphere. Would it be better to label this as Ttrop?

"Tsup" is a rather odd abbreviation for surface temperature.

Page 11 line 7: "explicit" should be "explicitly give"

Page 15 line 13 – "Selee" should be "Seele"

Page 17 line 8 – missing '('  ?????????

Figure 9 –"specttrocscopy" should be "spectroscopy"

---

## Author Comment (AC2) · 21 Nov 2017

**Replies to Reviewer 2**

We thank both reviewers for their knowledgeable and valuable comments. Our efforts in addressing them, together with the reviewers' suggestions, led to a revised manuscript that represents a great improvement with respect to the original version. In what follows, reviewer's comments are in black and authors' replies in red.

In general, the introduction in section 3 to 5 repeats the basics of radiative transfer, balancing instruments and optimal estimation, which are already described in detail in the cited literature. In my opinion, these sections could be easily shortened without loss of information.

We shortened the sections as suggested by the reviewer, also eliminating 13 basic equations.

The abstract (p.1 line 13) and section 3 describe the VESPA-22 back-end as an FFTS with 500 MHz bandwidth and 31kHz resolution. However, the schematic in Fig. 1(b) shows a system with 1GHz bandwidth. Is this the actual schematic of the instrument?

No, there was a mistake in the schematics. It has been corrected.

Also P4 line 1 states a 2GS/s sampling rate, which results in a 1 GHz Nyquist bandwidth.

The effective sampling rate is 1 GS/s. It was corrected in the revised manuscript.

P. 3 line 13: The sentence "the waveguide used by VESPA-22 polarizes the incoming radiation with a gain difference between the two polarization modes of 45 dB" is both incorrect and irrelevant. The rectangular waveguide supports only a single polarization by definition. The stated 45dB may refer to the feed horn, but the actual cross-polar with the offset reflector will be much worse. Bertagnolio 2012 states for the same instrument 35dB and 24dB, respectively. This should not affect the observations, so the sentence could be easily removed.

We agree with the reviewer and removed the sentence.

P. 6 line 28 states that the opacity in Fig 3 was calculated for water vapor profiles measured by AURA/MLS. However, the dominant effect in this figure is due to tropospheric water vapor which is not measured by MLS.

We clarified this statement in the revised manuscript. The profiles used were obtained merging tropospheric and stratospheric measurements collected by radiosondes from Eureka (80.0°N -85.9°W), Canada, and Aura/MLS, respectively.

P. 9 line 24 states that a 6 MHz wide interval around the line center is kept at 31kHz resolution, while all other channels are binned with 50 channels. In Fig 6(b) the interval without binning looks much smaller than 6MHz.

The interval without binning is indeed 6 MHz and was correctly showed in Figure 6b of the original manuscript. However, the x-axis unit in figure 6b was MHz and there was a  $10^4$  multiplicative factor on the bottom right of the figure. We realized that this indication was misleading and we changed the frequency unit for all the figures of the revised manuscript from MHz to GHz.

According to p. 14 line 1 the matrix with the measurement error Se was assumed with constant diagonal elements which are calculated from the residuals after an initial run of the retrieval. However, the central channels without binning will have a higher measurement error as the ones without binning. Please provide also the range of measurement errors which were used in Se for the daily retrievals. The retrieval errors in Fig 11 seem to be very small. Could this be an artifact of an underestimation of the measurement errors in Se due to the binning at the line wings?

The reviewer is correct and in the original manuscript Se was underestimated. In the new analysis inserted in the revised manuscript Se is now calculated before the smoothing process of the spectrum and this brought an increase of the retrieval uncertainty (now indicated as "spectral uncertainty") of approximately 70% over the entire vertical profile. The figure below (also inserted in the revised manuscript) shows the new error analysis where we also added the uncertainty due to the use of the second order polynomial (see later comment of this reviewer and a similar comment of reviewer 1).

The retrieval uncertainty (now "spectral uncertainty") is still small, especially in the lower stratosphere, but we underline that it represents only the uncertainty due to spectral noise and potential spectral artifacts.

The figure below shows the time series of the Se values. The range of Se values has been indicated in the revised manuscript.

---

## Author Response (AR2)

Dear Editor,

We addressed all the minor comments of the reviewer. In particular, we changed figure 4 and modified the text where his comments indicated it needed to be modified. Following one of his comments we also updated (very minor changes) figures 13, 14 and 15.

Kind regards,
Gabriele Mevi and Giovanni Muscari